# Heavy tails and pruning in programmable photonic circuits for universal unitaries

**Sunkyu Yu** [1] ✉ **& Namkyoo Park** [2] ✉

Developing hardware for high-dimensional unitary operators plays a vital role in implementing quantum computations and deep learning accelerations. Programmable photonic circuits are singularly promising candidates for universal unitaries owing to intrinsic unitarity, ultrafast tunability and energy efficiency of photonic platforms. Nonetheless, when the scale of a photonic circuit increases, the effects of noise on the fidelity of quantum operators and deep learning weight matrices become more severe. Here we demonstrate a nontrivial stochastic nature of large-scale programmable photonic circuits—heavy-tailed distributions of rotation operators—that enables the development of high-fidelity universal unitaries through designed pruning of superfluous rotations. The power law and the Pareto principle for the conventional architecture of programmable photonic circuits are revealed with the presence of hub phase shifters, allowing for the application of network pruning to the design of photonic hardware. For the Clements design of programmable photonic circuits, we extract a universal architecture for pruning random unitary matrices and prove that "the bad is sometimes better to be removed" to achieve high fidelity and energy efficiency. This result lowers the hurdle for high fidelity in large-scale quantum computing and photonic deep learning accelerators.

A unitary operation is an essential building block of quantum[1–3] and classical[4,5] linear systems because any linear operator can be decomposed into a set of unitary and diagonal operators[6]. With advances in quantum computations[3] and deep learning accelerators[7], development of reconfigurable hardware for universal unitary operations has become a topic of intense study. A programmable photonic circuit is one of the most widely used platforms[8,9] for unitary operations in optical neural networks[4,10], modal decoding[5] and quantum computations[1–3].

The fundamental strategy to realize universal unitary operators is to factorize the target operator of the $n$-degree unitary group U($n$) into the diagonal operators and unitary operators of the lower-degree group, such as SU(2)[11]. These subsystems can be realized with conventional optical elements, such as beam splitters, Mach–Zehnder

interferometers (MZIs) and phase shifters, which constitute a programmable photonic circuit with reconfigurable modulation. Although the mesh composed of these unit elements can perform universal unitary operations, the connectivity inside the mesh is non-unique and involves an optimal design issue for more compact and robust platforms[12–15]. To improve the original proposal—the Reck design[11]—for the mesh topology, recent approaches have successfully demonstrated advanced arrangements of two-channel subsystems—the Clements design[12]—and the advantages of utilizing multichannel building blocks—the Saygin design[13].

When each channel of the mesh is assigned as a node, a photonic circuit can be interpreted as a graph network[16] regardless of design strategy. Accordingly, it is logical to seek inspiration from network science[17] to understand and improve the large-scale mesh topology of

[1]Intelligent Wave Systems Laboratory, Department of Electrical and Computer Engineering, Seoul National University, Seoul 08826, Republic of Korea. [2]Photonic Systems Laboratory, Department of Electrical and Computer Engineering, Seoul National University, Seoul 08826, Republic of Korea. ✉e-mail: sunkyu.yu@snu.ac.kr; nkpark@snu.ac.kr

high-degree unitary groups, which should inherit intriguing features of complex networks. In this context, one promising issue is the degree distribution describing the differentiated importance of network nodes, which has been a hot topic through the concepts of heavy-tailed distributions, hub nodes and scale-freeness[17–21]. When the multiple decomposition processes are applied to U($n$)[11,12], a natural question arises: Do every decomposition and corresponding optical element contribute equally to the designed unitary operation? The answer to this question is of fundamental and practical importance in quantum physics and photonics for devising more advanced hardware architecture applicable to universal quantum evolutions and deep learning accelerators, especially with large-scale photonic circuits. Very recently, the first step toward this answer was achieved in the Reck design with asymmetric architecture to devise a stable algorithm in the self-configuration for large-scale multiport interferometers[22].

In this paper, we reveal that some subsystems are more important than others, even in the highly symmetric architecture of large-scale programmable photonic circuits. By applying various statistical models to programmable photonic circuits targeting universal unitaries, we verify that a type of unit rotation operator has a heavy-tailed distribution. This finding shows the presence of hub optical elements and the Pareto principle in photonic circuits, which enables the development of the pruning technique[23] for linear quantum or classical hardware. We demonstrate that the suggested hardware pruning for random unitaries allows for improved fidelity when the elements with noise above a specific threshold are removed. This result provides a design strategy for high fidelity and energy efficiency in large-scale quantum computations and photonic deep learning accelerators.

## Results

### Rotation operators in programmable photonic circuits

Before applying the statistical analysis to large-scale programmable photonic circuits, we revisit the Clements design[12], which is one of the most widely used architectures for universal unitaries. Figure 1 shows a schematic of the photonic circuit for the $n \times n$ unitary matrix $U_n \in$ U($n$) obtained from the Clements design. Both the Reck and Clements

designs employ nulling the off-diagonal elements of $U_n$ by sequentially multiplying the programmable unit operations $T_m^l \in$ U($n$) ($1 \le m \le n - 1$, $1 \le l \le n$, $m$ and $l$ are integers). $T_m^l$ leads to the SU(2) operation on the Bloch sphere defined for the $m$th and ($m+1$)th channels to set the off-diagonal element ($l$, $m$) or ($m+1$, $l$) to be zero.

To maximally cover the SU(2) group with $T_m^l$, reconfigurable and independent control of the amplitude and phase differences between the $m$th and ($m+1$)th channels is necessary[8]. One of the most popular platforms for $T_m^l$ is to utilize two pairs of a stationary MZI and a tunable phase shifter in one arm[8,9,12], which involves two adjustable parameters of $\theta \in [0, \pi/2]$ and $\varphi \in [0, 2\pi]$ (Fig. 1a). While the phase shifts $\theta$ and $\varphi$ correspond to tunable $z$-axis rotations on the Bloch sphere, the stationary MZIs constitute the $-\pi/2$ $x$-axis rotations (Fig. 1b, c). The unit operator then becomes $T_m^l(\theta,\varphi) = R_x^m(-\pi/2)R_z^m(-2\theta)R_x^m(-\pi/2)R_z^m(-\varphi)$, where $R_a^m(\xi)$ is the $\xi$-rotation to the $a$-axis on the $m$·($m+1$) Bloch sphere, and $\theta$ and $\varphi$ are determined to satisfy nulling of the ($l$, $m$) or ($m+1$, $l$) element. The target unitary operator $U_n$ is reproduced with multiple $T_m^l$ operators and the remaining diagonal matrix $D_n$ after nulling, as follows:

$$U_n = D_n \left[ \prod_{\{m,l\} \in S_n} T_m^l(\theta_{m,l}, \varphi_{m,l}) \right] \quad (1)$$

where $S_n$ is the ordered sequence of $\{m, l\}$ pairs determined by the nulling process[12] and $D_n$ is realized with phase shifters (Fig. 1d). By newly defining $S_n$, the Clements design employs the highly symmetric arrangement of the MZIs (Fig. 1e), which decreases the device footprint by half and enhances robustness to optical losses compared to the Reck design (see Supplementary Note S1 for the detailed processes).

The reconfigurability for universal unitary operators is thus realized with the $z$-axis rotation $R_z$ obtained from tunable phase shifts $\theta$ and $\varphi$. As programmable devices, the noise and power consumption of photonic circuits are determined by the performance of modulating optical refractive indices $\Delta n$ in the phase shifters and the following changes of $\theta$ and $\varphi$, as $-L\Delta n$, where $L$ is the modulation length.

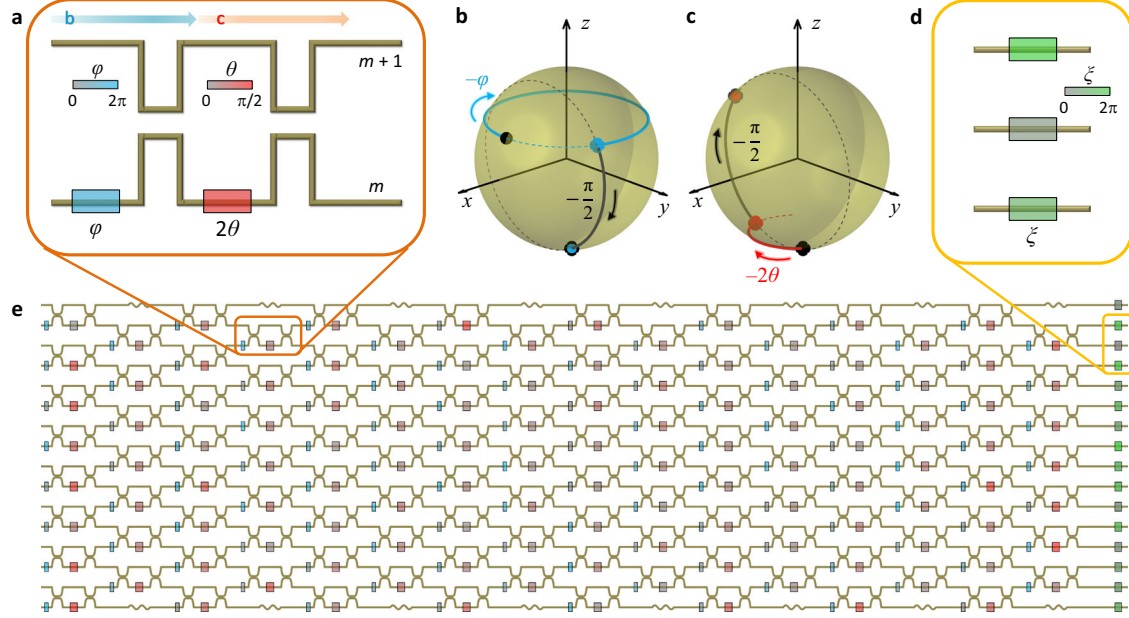

**Fig. 1 | Programmable photonic circuits for universal unitary operators.**
**a** Programmable photonic building block of $T_m^l$ composed of MZIs and phase shifters for the SU(2) operation between the $m$th and ($m+1$)th channels. Red and blue boxes represent the phase shifters for $\theta$ and $\varphi$, respectively. **b**, **c** The rotation operators of $R_x^m(-\pi/2)R_z^m(-\varphi)$ (**b**) and $R_x^m(-\pi/2)R_z^m(-2\theta)$ (**c**), described in Bloch spheres. Black and colored solid lines indicate $x$-axis and $z$-axis rotations, respectively. **b** and **c** correspond to the parts indicated by blue and red arrows in (**a**), respectively. **d** Phase shifters for the diagonal components of $D_n$. **e** Schematic diagram of the programmable photonic circuit for $U_{16}$. The tunability of $\theta$ and $\varphi$ allows for the programming of $U_{16}$.

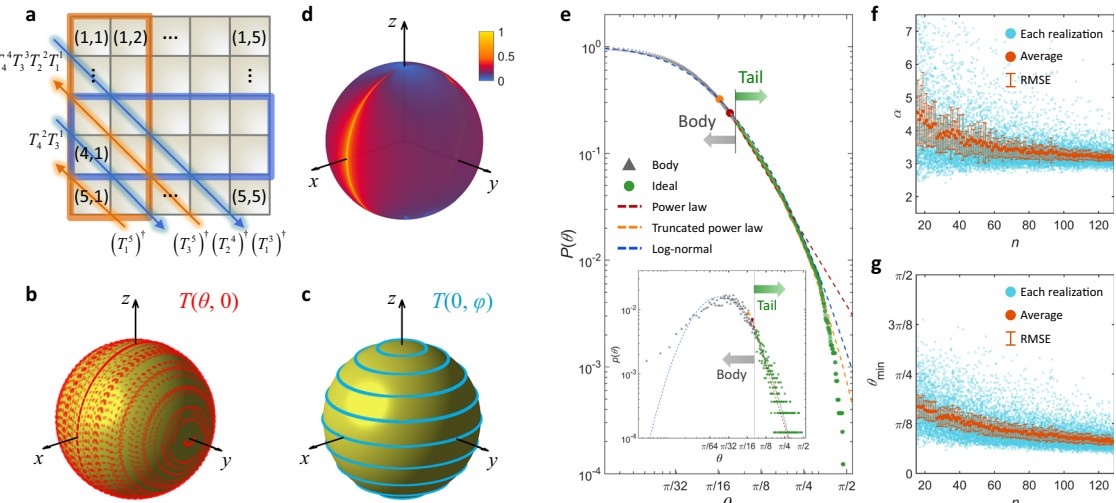

**Fig. 2 | Heavy-tailed distributions in unitary photonic circuits. a, b** Two origins of the heavy-tailed distributions of the rotation operators: unequal transformations in the nulling process (**a**) and nonuniform SU(2) rotations (**b**–**d**). **a** An example of the nulling process for $U_5$. Orange and green arrows denote the nulling of the off-diagonal elements with $UT^\dagger$ and $TU$, respectively. Red and blue boxes indicate the rotating components for the nulling of the (5,1) and (4,1) components, respectively. Rotated states with $T(\theta, 0)$ (**b**), $T(0, \varphi)$ (**c**), and $T(\theta, \varphi)$ (**d**). Each point in (**b**) and (**c**) denotes the transformed state through the corresponding $T$ applied to the uniformly random initial states on the Bloch sphere. The colors in the map in **d** depict the nonuniform density of the transformed states on the Bloch sphere. The initial states in (**b**) and (**c**) are obtained with 10 polar grids and 20 azimuthal grids (200

points), while 200 polar grids and 400 azimuthal grids (80,000 points) are used in (**d**). **e** Heavy-tailed distributions of $\theta$ described by the CCDF. The inset shows the PDF and its fitting. The body and tail are separated with $P(\theta) = 0.20$, referring to the Pareto principle. Red, orange and blue dashed lines show fitting with the power-law, power-law with an exponential cutoff (or truncated power-law) and log-normal distributions, respectively. Red and orange circles indicate the lower limit of the $\theta$-fitting for the power law and power law with an exponential cutoff, respectively. **f, g** The variations of the power-law estimators for different $n$: $\alpha$ (**f**) and $\theta_{min}$ (**g**). Each blue point represents a realization, and orange markers and error bars show the average and root-mean-square error (RMSE) of 100 random realizations at each $n$, respectively.

Therefore, the statistical analysis of the two adjustable phases $\theta$ and $\varphi$ is critical in examining the performance of large-scale programmable photonic circuits.

## Heavy tails in rotations

Due to the highly symmetric form of the photonic circuit (Fig. 1e), at first glance, it may appear to be reasonable to predict that the building blocks $T_m^l$ in the circuit have equal importance. Under this presumption, the distributions of $\theta$ and $\varphi$ should be statistically uniform for an ensemble of photonic circuits that generate random unitary operations uniformly distributed in U(n)[24]. Furthermore, it may also seem reasonable to expect similar distributions for $\theta$ and $\varphi$, both of which perform z-axis rotations.

However, upon closer inspection, we reveal that those presumptions are invalid. Instead, there are differences in the contributions of individual building blocks as well as the rotation operators of $\theta$ and $\varphi$. First, revisiting the nulling process of the Clements design[12], we note that each off-diagonal element of $U_n$ undergoes differentiated transformations. For example, in nulling the 5 × 5 unitary matrices (Fig. 2a), nulling the (5,1) and (4,1) components results in the $(T_1^5)^\dagger$-transformed 1st and 2nd columns and the $T_3^1$-transformed 3rd and 4th rows, respectively. Because the nulled off-diagonal elements no longer change, each building block treats a matrix element that undergoes a different number of SU(2) transformations; matrix elements that are nulled earlier get fewer transformations (see extended discussion in Supplementary Note S1).

These disparate transformations of each matrix element do not guarantee nontrivial distributions of the phase shifts $\theta$ and $\varphi$. However, the decomposed form of the building block operation $T_m^l(\theta, \varphi) = R_x^m(-\pi/2)R_z^m(-2\theta)R_x^m(-\pi/2)R_z^m(-\varphi)$ results in the nontrivial distribution of $\theta$, which is clearly distinct from that of $\varphi$. Figure 2b and c shows the transformations of the initial states uniformly distributed in the polar ($\xi$) and azimuthal ($\eta$) axes of the Bloch sphere by multiplying $T_m^l(\theta, \varphi = 0) = R_x^m(-\pi/2)R_z^m(-2\theta)R_x^m(-\pi/2)$ and $T_m^l(\theta = 0, \varphi) = R_x^m(-\pi)$

$R_z^m(-\varphi)$, respectively, where nonzero $\theta$ and $\varphi$ also have uniformly distributed values in their ranges. Notably, the transformed states by $T_m^l(\theta, \varphi = 0)$ become nonuniform (Fig. 2b), in sharp contrast to the uniform distribution from $T_m^l(\theta = 0, \varphi)$ (Fig. 2c). Such a discrepancy originates from the difference between the pure z-axis rotation $R_z^m(-\varphi)$ and the transformed rotation $R_x^m(-\pi/2)R_z^m(-2\theta)R_x^m(-\pi/2)$ and eventually leads to nonuniformity on the Bloch sphere for $T_m(\theta, \varphi)$ (Fig. 2d). We emphasize that the unequal contributions of each nulling (Fig. 2a) will accumulate the nonuniform distribution from the $\theta$ rotations, which leads to nontrivial statistics in the phase shift design.

To confirm this prediction, we investigate the statistics of $\theta$ and $\varphi$ in realizing programmable photonic circuits that reproduce random unitary operations achieved by uniformly sampling the U(n) group with the Haar measure[24]. We calculate the probability density functions (PDFs) $p(\theta)$ and $p(\varphi)$ and the complementary cumulative distribution functions (CCDFs) $P(\theta)$ and $P(\varphi)$ for an ensemble of 100 $U_n$ realizations at each $n$. As expected from the uniform distribution with $T_m^l(\theta = 0, \varphi)$ (Fig. 2c), the distribution of $p(\varphi)$ is trivially uniform (Supplementary Note S2).

One of the key findings of this work is the nonuniform distribution of $\theta$. Figure 2e shows an example of the $\theta$ distribution for $U_{128}$, which includes 8128 values. As shown in the linearized plots of the CCDF and PDF on the log-log scale, $\theta$ possesses a heavy-tailed distribution[17,19,21], indicating a smaller decrease in $p(\theta)$ for increasing $\theta$ than that of the exponential distribution. For the quantitative analysis, we employ three representative heavy-tailed distribution models[19]—power-law, power-law with an exponential cutoff and log-normal distributions—and the exponential distribution model. The models are fitted with the $\theta$ dataset of each realization of photonic circuits by utilizing analytical or numerical maximization of the model likelihoods[19,25] and the Kolmogorov–Smirnov test[26] for the models with lower bounds (see Methods for details). This standard procedure determines the range and shape of the tail of each model for the optimized fitting of a given dataset.

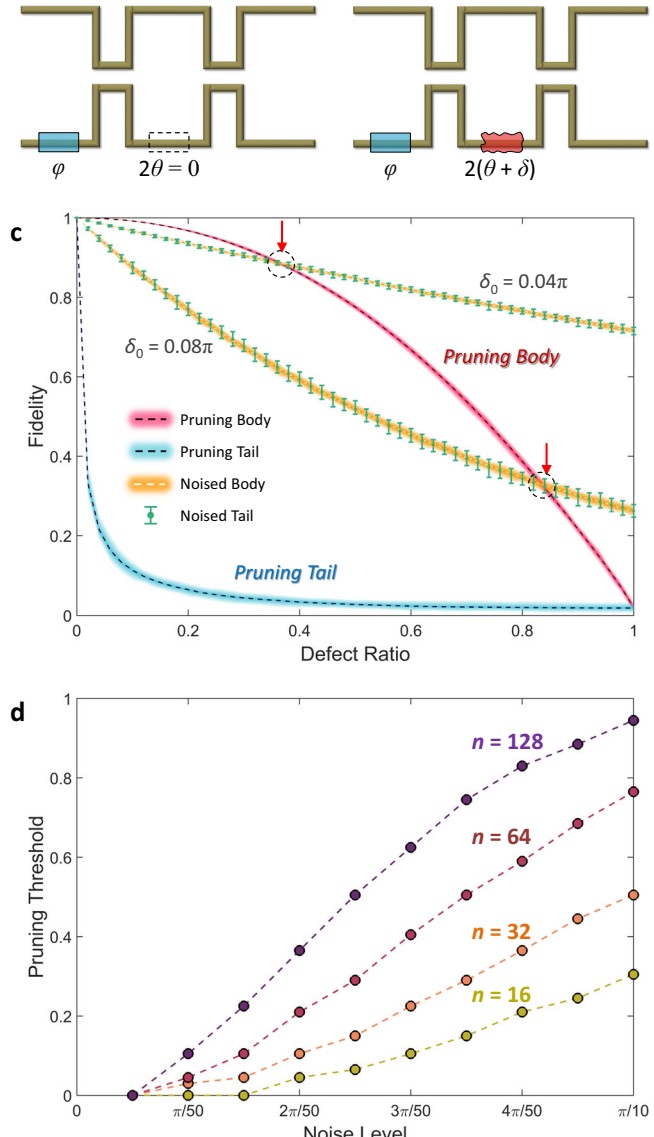

**Fig. 3 | Pruning is often better than noise. a** The concept of pruning in programmable photonic circuits. The phase shifter $2\theta$ of the building block is replaced with an ordinary waveguide, which preserves the symmetry in the MZI arms. **b** The noisy building block. The phase shifter $2\theta$ is perturbed as $2(\theta + \delta)$. **c** Comparison of the fidelities of the $U_{128}$ photonic circuits in different groups: pruning body (red line), pruning tail (blue line), noisy body (orange line) and noisy tail (green error bars). The thicknesses of the colored lines and the error bars present the range of the fidelities between their maxima and minima. The red arrows indicate the pruning thresholds for each case. Two pairs of groups with noisy bodies and noisy tails are shown for $\delta_0 = 0.04\pi$ and $0.08\pi$, which correspond to about 2K–4K temperature changes in silicon infrared thermo-optical phase shifters[27]. **d** Pruning threshold as a function of the noise level $\delta_0$ for different degrees of unitary operators. In (**c**) and (**d**), 100 random $U_n$ realizations are analyzed per value of $n$ and defect ratio.

Notably, all the heavy-tailed models provide good fits for large $n$, showing the consistent behaviors of their estimators for each realization, which is a critical condition for model consistency[21]. For example, the exponent $\alpha$ (Fig. 2f) and the lower bound $\theta_{min}$ (Fig. 2g) in the power-law model $P(\theta) = (\theta/\theta_{min})^{-\alpha+1}$ converge with increasing $n$, which demonstrates that the heavy-tailed distribution becomes more apparent in larger-scale programmable photonic circuits. The average of the power-law exponents at $n = 128$ is $\alpha_{avg} = 3.18$ for 100 realizations

(or 812,800 values of $\theta$), while lower and upper limits are $\alpha_{min} = 2.75$ and $\alpha_{max} = 3.78$. Such consistency clearly proves the validity of the power-law model[20,21] for describing the distribution of the $\theta$-rotation operators (Supplementary Notes S3–S5 for the results of the crossover heavy-tailed distributions and exponential distribution). We note that the averaged lower bound $\theta_{min} = 0.08\pi$ with $P(\theta_{min}) = 0.24$ shows that most of the significant rotations $0.08\pi \leq \theta \leq 0.50\pi$ come from ~24% of the building blocks, which illustrates the Pareto principle for large-scale programmable photonic circuits.

## Hub units and pruning

The observed heavy-tailed distribution of $\theta$ rotation operators signifies that some building blocks $T_m^l$ with $\theta$ in the 'Tail' part in Fig. 1e are more critical than other building blocks ('Body' part in Fig. 1e). In realizing programmable photonic circuits for universal unitary operations (Fig. 1a), many phase shifters in the "body" of the distribution may be unnecessary because $\theta \sim 0$. On the other hand, the "tail" phase shifters with large $\theta$ values operate as hub units. Because such hub units deliver most of the necessary $\theta$-rotations for realizing $U_n$, we can envisage the application of the pruning technique in computer science[23] to photonic hardware.

Figure 3a shows the concept of pruning for programmable photonic circuits. The entire photonic circuit for $U_n$ includes $n(n-1)/2$ number of SU(2) building blocks and the same number of $\theta$ values. We define the set of sorted $\theta$ values for a given photonic circuit as $\Theta_n = \{\theta_r | 1 \leq r \leq n(n-1)/2$ for the integer $r$ that denotes the index of each building block according to an order of $\theta$, as $\theta_p \leq \theta_q$ for $p \leq q\}$, where $\theta_r$ with larger $r$ represents a more important building block. The pruning of less important ones—body elements—for the photonic circuit is then defined by imposing $\theta_r = 0$ for $1 \leq r \leq \sigma$, where the integer $\sigma$ determines the degree of pruning: $\sigma = 0$ for preserving the original circuit and $\sigma = n(n-1)/2$ for entirely removing $\theta$ rotations in the circuit. In the hardware implementation, pruning corresponds to leaving out the phase shifters for $\theta$ and preserving the symmetry in the MZI arms.

The refractive index modulation in the phase shifters is responsible for much of the energy consumption and noise generation in programmable photonic circuits[8,9]. For example, consider a typical thermo-optic phase shifter with a device length of 100 μm[27], which operates at the telecom wavelength of 1550 nm and is based on silicon photonics technology. The amount of thermal noise present in the phase evolution is determined by the thermo-optic coefficient of silicon[28] $dn/dT = 1.8 \times 10^{-4} K^{-1}$, which can approach $0.02\pi$ per kelvin. This noise may be further exacerbated in larger-scale devices due to increasing thermal crosstalk. Therefore, pruning of superfluous phase shifters allows for more energy-efficient and noise-tolerant photonic circuits for reconfigurable unitary operations, provided that the circuit after the pruning accurately reproduces unitary operations. To examine the performance of pruning in a practical situation, we prepare three control groups: one group with the pruning of more important building blocks—tail elements—with $\theta_r = 0$ for $n(n-1)/2 - \sigma + 1 \leq r \leq n(n-1)/2$, and two groups with noisy elements. For the noisy elements, we assume random noise from the phase shifter by assigning the noise $\delta_k$ to the $k$th original rotation as $\theta_k + \delta_k$, where $\delta_k = u[0, \delta_0]$ represents the uniform random distribution between 0 and $\delta_0$. For a fair comparison, we construct the groups of noisy elements by replacing the body- or tail-pruned elements in the pruning groups with noisy elements.

To characterize the precision of the operation of the circuits with pruning or noises, we define the fidelity that quantifies the metric between the original and defective operators[29] as follows (see Methods for the derivation):

$$F(U_n^D, U_n^O) = \frac{2\text{Re}(\text{Tr}[(U_n^D)^\dagger U_n^O])}{n + \text{Tr}((U_n^D)^\dagger U_n^D)}, \quad (2)$$

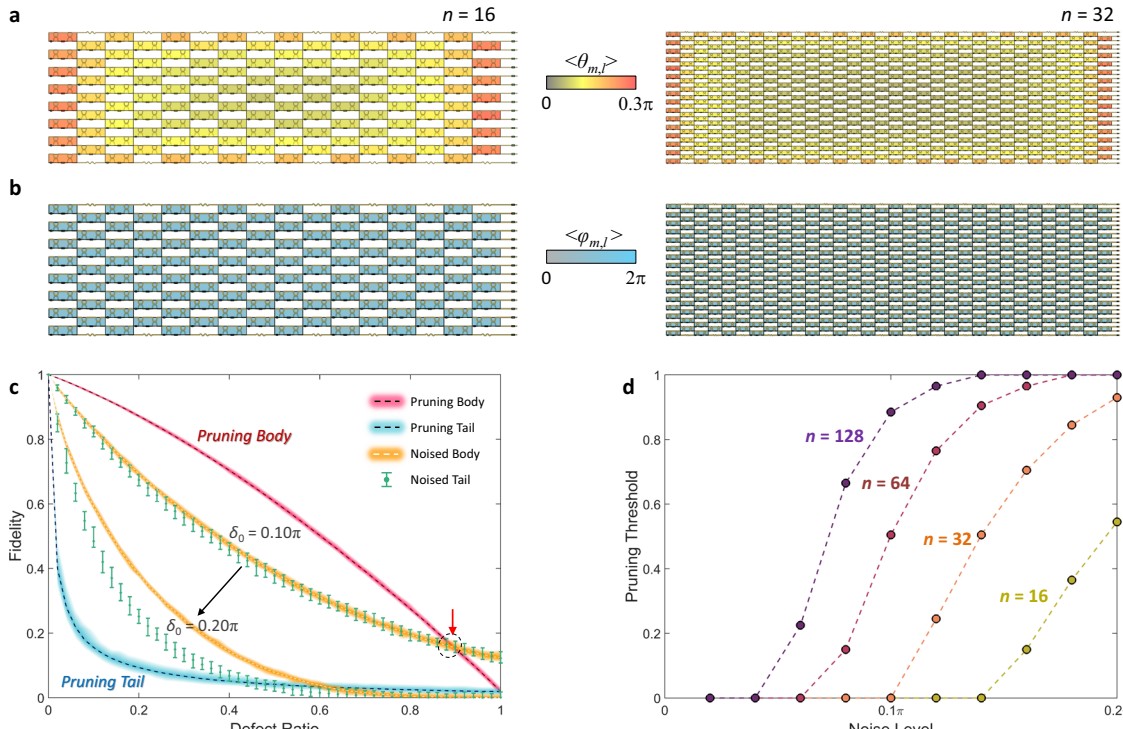

**Fig. 4 | Universal architecture for pruning in reconfigurable unitaries. a, b** The averages of $<\theta_{m,l}>$ (**a**) and $<\varphi_{m,l}>$ (**b**) for the photonic circuits of 100 $U_n$ realizations with $n = 16$ and $n = 32$. We set the upper bound of the color map in (**a**) to be $0.3\pi$ for better visibility. **c** Comparison of the fidelities of the $U_{128}$ photonic circuits in different groups: pruning body (red line), pruning tail (blue line), noisy body (orange line) and noisy tail (green error bars). The thicknesses of the colored lines and the error bars present the range of the fidelities between their maxima and minima. The red arrow indicates the pruning threshold. Two pairs of groups with noisy bodies and noisy tails are shown for $\delta_0 = 0.10\pi$ and $0.20\pi$, which correspond to about 5K-10K temperature changes in silicon infrared thermo-optical phase shifters[27]. **d** Pruning threshold as a function of the noise level $\delta_0$ for different degrees of unitary operators.

where $U_n^O$ and $U_n^D$ represent the original unitary matrix and its defective (pruned or noisy) one, respectively, and Tr(*A*) is the trace of the square matrix *A*. Figure 3c shows the fidelities of each photonic circuit with the pruning or noise as a function of the ratio of defective elements: $2\sigma/n(n-1)$ in the pruning groups. As expected, the fidelity is preserved much better when the body is pruned instead of the tail. More critical results are shown in comparison with the noisy circuits. When the noise amplitude increases, removing a specific ratio of the "body" phase shifters can be better for higher fidelity than the noisy ones, whether the noise is imposed on body or tail elements. Such a ratio, called the pruning threshold, increases with the noise level and scale of photonic circuits (Fig. 3d). This result states that there is a substantial restriction on the noise level in a large-scale programmable photonic circuit. If a phase shifter cannot meet this restriction, then it is better to remove the phase shifter to increase accuracy and decrease energy consumption for reconfigurability.

## Universal architecture for pruning

Although the result shown in Fig. 3 demonstrates hub functionality and the advantage of pruning in realizing an individual unitary operator, it is insufficient to apply pruning to programmable photonic circuits for universal unitary operators. This is because the sorted set $\Theta_n$ for pruning varies with the form of a unitary operator. To apply the pruning method for universal unitaries with reconfigurability, it is necessary to construct an adaptable architecture for the pruning process.

Because the position of each building block for nulling a specific off-diagonal element is fixed in an *n*-degree photonic circuit, the averages of the phase rotations $<\theta_{m,l}>$ and $<\varphi_{m,l}>$ are well-defined in hardware for random unitary operations that are uniformly sampled from U(*n*). Figure 4a and b describes the universal architectures defined by $<\theta_{m,l}>$ and $<\varphi_{m,l}>$, respectively, for 100 $U_n$ realizations of

$n = 16$ and $n = 32$ (see Supplementary Note S6 for $n = 64$). As expected from the distinct SU(2) operations from $\theta$ and $\varphi$ (Fig. 2b, c), we observe a spatially inhomogeneous distribution of $<\theta_{m,l}>$ in contrast to that of $<\varphi_{m,l}>$. More specifically, the universal architectures show significant $\theta$-rotation contributions from the building blocks near the boundary of the programmable photonic circuits. Such a consistent distribution allows for a universal sorted set $<\Theta>_n = \{<\theta_{m,l}>_r | 1 \leq r \leq n(n-1)/2$ for the integer *r* that denotes the index of each building block according to an order of $<\theta_{m,l}>$, as $<\theta_{m,l}>_p \leq <\theta_{m,l}>_q$ for $p \leq q\}$ to develop a pruning process applicable to any unitary operations.

From this guideline, we again employ pruning and add noise to the body and tail elements to the set $<\Theta>_n$. As shown in Fig. 4c and 4d, the general tendencies in Fig. 3c and 3d are preserved; the tail is more important than the body, the bad is better to be removed and pruning is more efficient for larger-scale photonic circuits. Although the minimum noise level increases, there is still a pruning threshold that guarantees the advantage of removing $\theta$ phase shifters, and this tendency is much more apparent in larger-scale programmable photonic circuits. Notably, the importance of protecting hub elements from noise becomes evident at the strong noise level ($\delta_0 = 0.20\pi$ cases in Fig. 4c). Furthermore, let us consider one of the state-of-the-art realizations of experimentally demonstrated programmable photonic circuits, which allows for $n = 64$ matrix multiplications[30,31]. The realization then requires 2016 (=$n(n-1)/2$) unit cells composed of 4032 MZIs and 2016 $\theta$- and $\varphi$-phase shifters for unitary operations. When we consider thermal noise from about 5K temperature change, Fig. 4d shows that it is more advantageous to remove 50% (or 1008) $\theta$-phase shifters in realizing U(64) circuits.

## Pruning in photonic deep neural networks

The importance of achieving high-fidelity photonic U(*n*) operations demonstrated in Fig. 4 has been widely recognized in quantum

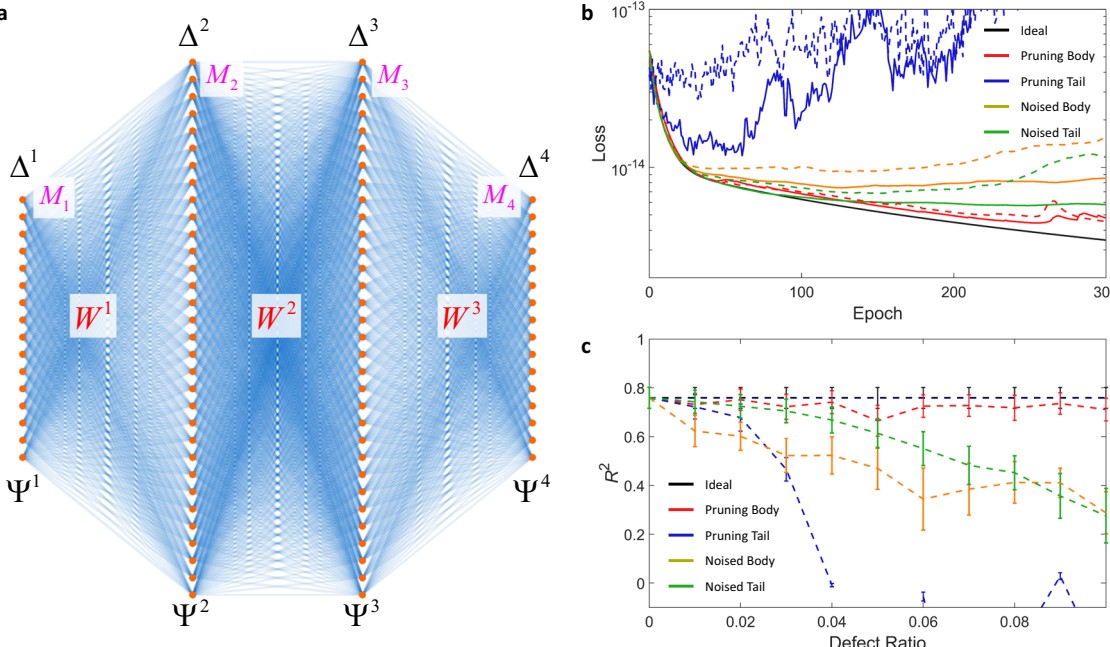

**Fig. 5 | Pruning in photonic deep neural networks for regression. a** The architecture of the deep neural network for analyzing the effect of pruning. $M_p$ ($p = 1, 2, 3, 4$) is the number of neurons in the $p$th layer, where $M_1 = M_4 = 16$ and $M_2 = M_3 = 32$. $\Psi^p = [a_1{}^p, a_2{}^p, \ldots, a_{Mp}{}^p]^T$ and $\Delta^p = [\delta_1{}^p, \delta_2{}^p, \ldots, \delta_{Mp}{}^p]^T$ are the signal and error column vectors at the $p$th layer, respectively, where $a_k{}^p$ and $\delta_k{}^p$ are the signal and error at the $k$th neuron of each layer ($k = 1, 2, \ldots, M_p$), respectively. $W^p$ is the weight matrix between the $p$th and ($p+1$)th layers. **b, c** Learning curves (**b**) and R-squared accuracies (**c**) of the model, which are estimated with the test dataset. The colors of the lines denote the unitary photonic circuits in different groups: pruning body

(red), pruning tail (blue), noisy body (orange) and noisy tail (green). Noisy body and noisy tail are shown for $\delta_O = 0.001\pi$. In (**b**), solid and dashed lines denote 5% ($2\sigma/n(n-1) = 0.05$) and 10% ($2\sigma/n(n-1) = 0.10$) defect ratios, respectively. In **c**, dashed lines and error bars represent the mean value and half of the standard deviation of the test dataset accuracies of 1000 realizations. The disconnection of the blue dashed line in (**c**) denotes the failure of the training due to the divergence of network parameters. All the other parameters for training and calculating loss functions are shown in Methods.

computing, such as boson sampling[1] or quantum Fourier transform[32] in linear-optical quantum computing[33]. On the other hand, the validity of pruning in deep learning requires further clarification because unitary operations are part of an entire neural network composed of weight matrices and activation functions[4,34]. Therefore, we analyze how the pruning and adding noise in the unitary subparts of weight matrices impact the performance of photonic deep neural networks[4].

To focus on the effect of pruning and adding noise, we consider a traditional feedforward neural network[34,35] (Fig. 5a) with conventional training[36] and evaluation methods. The target problem is the regression of the relationship between the input and output datasets, which are connected through the neural network of Fig. 5a with a specific set of weight matrices $\{W^p | p = 1, 2, \text{ and } 3\}$. The goal of the network training starting from initially random weight matrices is the inference of the test output dataset from the test input, which corresponds to finding a specific form of a deep neural network from random deep neural networks (see Methods for details of the neural network model: architecture, datasets, training, hyperparameters and loss function).

When employing programmable photonic circuits to deep neural networks, the weight matrix $W = W^p$ is realized through the singular value decomposition[6] $W = UDV^\dagger$, where $U$ and $V^\dagger$ are unitary matrices and $D$ is a diagonal matrix[4,8]. The unitary- and diagonal-matrix operations can be implemented with the structures in Fig. 1e and Fig. 1d, respectively[8], while gain or loss may be necessary for diagonal operations. Among those sub-operations of $W$, we apply pruning or adding noise to the unitary matrices $U$ and $V^\dagger$ through the procedure in Fig. 4. Each weight matrix is then replaced with the defective one: $W^D = U^D D V^{D\dagger}$, where $U^D$ and $V^{D\dagger}$ are the pruned or noisy unitary matrices.

Figure 5b and c demonstrates the impact of pruning in the regression problem. The learning curves estimated with the test

dataset (Fig. 5b) show that the pruning body (red solid and dashed lines) allows for much more robust network learning than all the other cases of defective unitaries. Noisy cases present relatively unstable learning, especially when the ratio of defective elements increases (solid lines for 5% and dashed lines for 10% defective elements). Notably, the pruning tail—removing the operations of hub elements—results in the complete failure of learning. Such a discrepancy between pruning and adding noise is also apparent in the R-squared regression accuracy ($R^2$) estimated after 300 epochs of training (Fig. 5c), which shows that only the pruning body case provides an accuracy close to the ideal case without any defects (black solid line). Although the case examined in Fig. 5b, c is a specific regression problem of a deep neural network model, the result shown in the same vein as Fig. 4 demonstrates the validity of the pruning method in photonic deep neural networks, serving as proof-of-concept at least. To ensure reproducibility, we include the codes for Figs. 2–5 in Supplementary Code 1.

## Discussion

Due to the mathematical generality of our study, the presented results should be universal for programmable photonic[8,9] or superconducting[37] processors for reconfigurable unitary operations when the unit SU(2) operation is nonuniform on the Bloch sphere and the target degree $n$ is finite. Notably, we observed the excellent fitting with the power-law model, the crossover behaviors from exponential to heavy tails in the truncated power-law and log-normal models, and the evident failure of the exponential model, nearly above the degree $n \geq 80$. It is worth mentioning that the state-of-the-art realization of programmable photonic circuits using MZIs and phase shifters allows the $n = 64$ matrix multiplication[30,31], which is close to the heavy-tailed regime. Therefore, the heavy-tailed features are evident at the scale near and beyond the present

state-of-the-art degrees ($n$-$10^2$) in deep learning accelerators[4,38,39] and noisy intermediate-scale quantum computers[3,40,41]. The suitable application of the demonstrated pruning method, which allows for leaving out a significant portion of electro-optic modulations in programmable photonic circuits, will become particularly beneficial for the next era of quantum computing and deep learning hardware.

Although we studied the performance of pruning in photonic deep neural networks, various issues still remain for practical realizations. First, although we focused on unwanted phase shifts that may originate from thermal crosstalk, optical absorption from material loss or imperfect device fabrication may be worse in large-scale photonic neural networks, especially when using gain or loss media[42]. Second, the effect of other forms of nonlinear activation functions[43] and network architectures[44] on pruning performance should be studied. Finally, the statistical properties of weight matrices depend on problems, model architectures, learning methods and hyperparameters[45,46]. The resulting unitary matrices can have some biased distributions distinct from random Haar matrices, which will impose the problem- or model-specific properties in pruning performance.

The presence of heavy-tailed distributions in programmable photonic circuits inspires the extension of seminal achievements in probability theory and network science to wave physics. As shown in our study, the intriguing features related to heavy-tailed distributions are demonstrated in wave platforms, such as the observed Pareto principle in wave physics and the critical role of hub elements in pruning and noise immunity. Regarding the emergence of heavy-tailed distributions in network science[18], the phase shift $\theta$, or the rotation $R_z^m(-2\theta)$, may correspond to the number or weight degree of links that network nodes possess. In this analogy, each SU(2) unit corresponds to a network node, and different design methodologies of the U($n$) decomposition[11–13] imply a set of distinct network architectures for the same signal behavior U($n$). To complete this analogy between wave physics and network science, we can envisage a network metric that quantifies the connectivity between SU(2) units, which should consider the interference effect, as demonstrated in the network modeling of wave scattering systems[47]. When the connectivity of integrated wave systems becomes more extensive and complex[48], the concepts of complex networks will provide a foundation for design strategies in wave physics.

In conclusion, we demonstrated that some of the unit elements in a large-scale programmable photonic circuit are more important than others, exhibiting the heavy-tailed feature verified with conventional statistical models, i.e., the power-law, power-law with an exponential cutoff and log-normal distributions, and the exponential distribution as a counterexample. The observed heavy-tailed distribution originates from nonuniform rotations on the Bloch sphere, which are ubiquitous in conventional SU(2) units for programmable photonic circuits. The result allows for the design strategy—pruning—for high fidelity and energy efficiency, which offers intriguing insight into the design of large-scale photonic structures for classical and quantum devices, as demonstrated in the application to photonic deep neural networks. Further research on devising other forms of SU(2) units or the units with higher degree for $U_n$ factorization is desirable to alter the observed heavy-tailed distributions.

## Methods
### Model fitting process
To analyze the $\theta$ distributions in an ensemble of programmable photonic circuit realizations, we employ multiple statistical models: power-law, power-law with an exponential cutoff, log-normal and exponential distributions. Each model is defined by a set of model parameters $\{q_s\}$. To calculate the model parameters for the fitting of a given dataset $\{\theta_1, \theta_2, ..., \theta_M\}$, we employ an analytical or numerical calculation of the maximum likelihood estimators (MLEs)[25] from the PDF $p(\theta_m; \{q_s\})$, which defines the probability of finding $\theta_m$ with the model having the parameters $\{q_s\}$. First, the probability of obtaining the dataset from the statistical model with the given model parameters $\{q_s\}$ and the PDF $p(\theta_m; \{q_s\})$ is

$$p(\{\theta_m\}; \{q_s\}) = \prod_{m=1}^{M} p(\theta_m; \{q_s\}), \tag{3}$$

which is called the likelihood for the data and model. The model with the higher likelihood then provides the better fit to the dataset $\{\theta_m\}$[19]. Because the employed statistical models have exponential forms, it is conventional to utilize the log-likelihood $L$:

$$L = \sum_{m=1}^{M} \log(p(\theta_m; \{q_s\})). \tag{4}$$

The fitting of the model to a given set of data, which requires the calculation of $\{q_s\}$, then corresponds to the maximization of $L$ with respect to $\{q_s\}$. Therefore, the MLE is defined as

$$\nabla_{\{q_s\}} L = \nabla_{\{q_s\}} \sum_{m=1}^{M} \log(p(\theta_m; \{q_s\})) = O. \tag{5}$$

### Power-law distribution model
In analyzing the heavy-tailed statistics of the $\theta$-rotations, we mainly employ the power-law distribution model[17–19], which supports the PDF and CCDF, as follows:

$$p(\theta) = \frac{\alpha - 1}{\theta_{min}} \left( \frac{\theta}{\theta_{min}} \right)^{-\alpha}, \tag{6}$$

$$P(\theta) = \left( \frac{\theta}{\theta_{min}} \right)^{-\alpha + 1}, \tag{7}$$

where $\alpha$ and $\theta_{min}$ are the exponent and lower bound of the power-law model, respectively. The model is defined in the range $\alpha > 1$, and the model parameter set is $\{q_s\} = \{\alpha\}$. For a given dataset, the log-likelihood becomes

$$L = M \log(\alpha - 1) + M(\alpha - 1) \log \theta_{min} - \alpha \sum_{m=1}^{M} \log \theta_m. \tag{8}$$

The MLE then leads to $\alpha$, as

$$\alpha = 1 + M \left[ \sum_{m=1}^{M} \log \left( \frac{\theta_m}{\theta_{min}} \right) \right]^{-1}. \tag{9}$$

We calculate an array of $\alpha$ values using Eq. (9) for all the possible values of $\theta_{min}$, where each pair of $\alpha$ and $\theta_{min}$ comprises a candidate power-law model.

### Power-law model with an exponential cutoff
To obtain a thorough confirmation of the heavy-tailed statistics, we test crossover distributions between a power-law and an exponential distribution. First, we apply the power-law model with an exponential cutoff, which is the truncated version of the original power-law model.

The PDF and CCDF of the model are[17,19]:

$$p(\theta) = \frac{\lambda_c^{1-\alpha_c}}{\Gamma(1-\alpha_c, \lambda_c\theta_{c,\min})}\theta^{-\alpha_c}e^{-\lambda_c\theta}, \qquad (10)$$

$$P(\theta) = \frac{\Gamma(1-\alpha_c, \lambda_c\theta)}{\Gamma(1-\alpha_c, \lambda_c\theta_{c,\min})}, \qquad (11)$$

where $\alpha_c$, $\lambda_c$, and $\theta_{c,\min}$ are the power-law exponent, cutoff exponent and the lower bound of the model, respectively, and $\Gamma(s,x)$ is the upper incomplete gamma function. The model is defined in the range of $\alpha_c \geq 0$ and $\lambda_c \geq 0$. The log-likelihood for the dataset $\{\theta_1, \theta_2, ..., \theta_M\}$ is

$$L = M(1-\alpha_c)\log\lambda_c - M\log\Gamma(1-\alpha_c, \lambda_c\theta_{c,\min}) - \alpha_c\sum_{m=1}^{M}\log\theta_m - \lambda_c\sum_{m=1}^{M}\theta_m. \qquad (12)$$

Although the MLE with the model parameters $\{q_s\} = \{\alpha_c, \lambda_c\}$ leads to the following relations:

$$\log\lambda_c + \frac{\partial_{\alpha_c}\Gamma(1-\alpha_c, \lambda_c\theta_{c,\min})}{\Gamma(1-\alpha_c, \lambda_c\theta_{c,\min})} = -\frac{1}{M}\sum_{m=1}^{M}\log\theta_m,$$
$$\frac{1-\alpha_c}{\lambda_c} + \frac{\theta_{c,\min}(\lambda_c\theta_{c,\min})^{-\alpha_c}e^{-\lambda_c\theta_{c,\min}}}{\Gamma(1-\alpha_c, \lambda_c\theta_{c,\min})} = \frac{1}{M}\sum_{m=1}^{M}\theta_m, \qquad (13)$$

we instead employ the numerical minimization of $-L$ with the constraints $\alpha_c \geq 0$ and $\lambda_c \geq 0$ due to the difficulty in handling the analytical derivative of the upper incomplete gamma function. We calculate the pairs of $\alpha_c$ and $\lambda_c$ for all the possible values of $\theta_{c,\min}$, where a set of $\alpha_c$, $\lambda_c$, and $\theta_{c,\min}$ comprises a candidate for the model.

### Log-normal distribution model

To cover the intermediate regime between the power-law and exponential distributions[17], we employ another crossover distribution: the log-normal distribution model. The PDF and CCDF of the model are[17,19]:

$$p(\theta) = \frac{1}{\sigma\theta\sqrt{2\pi}}\exp\left(-\frac{(\log\theta - \mu)^2}{2\sigma^2}\right), \qquad (14)$$

$$P(\theta) = \frac{1}{2}\left[1 - \text{erf}\left(\frac{\log\theta - \mu}{\sigma\sqrt{2}}\right)\right], \qquad (15)$$

where $\mu$ and $\sigma$ are the mean and standard deviation of $\log(\theta)$, respectively, and erf is the error function. With the model parameters $\{q_s\} = \{\mu, \sigma\}$, the log-likelihood and the MLE relation are shown in Eqs. (16) and (17), respectively, as follows:

$$L = -\sum_{m=1}^{M}\log\theta_m - M\log\sigma - \frac{M}{2}\log 2\pi - \sum_{m=1}^{M}\frac{(\log\theta_m - \mu)^2}{2\sigma^2}. \qquad (16)$$

$$\sum_{m=1}^{M}\frac{\log\theta_m - \mu}{\sigma^2} = 0, \frac{M}{\sigma} = \sum_{m=1}^{M}\frac{(\log\theta_m - \mu)^2}{\sigma^3}. \qquad (17)$$

Instead of utilizing the analytical MLE, we employ numerical minimization of $-L$ with the constraint $\sigma \geq 0$.

### Exponential distribution model

For the comparison with models other than heavy-tailed distributions, we test the exponential distribution model[17,19,21], which has the

following PDF and CCDF:

$$p(\theta) = \lambda_e e^{\lambda_e\theta_{e,\min}}e^{-\lambda_e\theta}, \qquad (18)$$

$$P(\theta) = e^{\lambda_e\theta_{e,\min}}e^{-\lambda_e\theta}, \qquad (19)$$

where the model parameter is $\{q_s\} = \{\lambda_e\}$. The log-likelihood and the MLE relation are

$$L = \log\lambda_e + M\lambda_e\theta_{e,\min} - \lambda_e\sum_{m=1}^{M}\theta_m, \qquad (20)$$

$$\lambda_e = \left[\sum_{m=1}^{M}\theta_m - M\theta_{e,\min}\right]^{-1}. \qquad (21)$$

We calculate an array of $\lambda_e$ values using Eq. (21) for all the possible values of $\theta_{e,\min}$, where each pair of $\lambda_e$ and $\theta_{e,\min}$ comprises a candidate for the model.

### Kolmogorov–Smirnov test

In the power-law, power-law with an exponential cutoff and exponential distribution models, we obtain multiple candidates for the models with different values of lower bounds $\theta_{\min}$, $\theta_{c,\min}$ and $\theta_{e,\min}$, respectively. Each candidate of a model supports a distinct range of data for model validity and possesses different values of model parameters $\{q_s\}$. To extract the optimum model among the candidates, we apply the Kolmogorov–Smirnov (KS) test[19,26]. When the CDFs of the dataset and the statistical model are $S(\theta)$ and $P(\theta; \theta_{\min}, \{q_s\})$ for the lower bound parameter $\theta_{\min}$, we define the maximum distance $D$ between the data and model distributions as:

$$D = \max_{\theta \geq \theta_{\min}}|S(\theta) - P(\theta; \theta_{\min}, \{q_s\})|. \qquad (22)$$

We select $\theta_{\min}$ and the corresponding $\{q_s\}$ to minimize $D$, determining the optimum statistical model for each case of the power-law, power-law with an exponential cutoff and exponential distribution models. This optimum model has the tail for the best fitting of a given dataset within the definition of each distribution.

### Fidelity for unitary matrices

We consider the $n \times n$ unitary matrix $U_n^O$ and its defective one $U_n^D$, which could be nonunitary in general. The cost function or the square of the metric between the matrices is defined by[29]:

$$J_U = \frac{1}{n^2}\sum_{i,j}|U_n^O{}_{(i,j)} - U_n^D{}_{(i,j)}|^2$$
$$= \frac{1}{n} + \frac{1}{n^2}\text{Tr}((U_n^D)^\dagger U_n^D - 2\text{Re}[(U_n^D)^\dagger U_n^O]), \qquad (23)$$

where $A_{(i,j)}$ is the $(i,j)$ matrix component and $\text{Tr}(A)$ is the trace of the square matrix $A$. Because $J_U \geq 0$, we obtain the relationship:

$$n + \text{Tr}((U_n^D)^\dagger U_n^D) \geq 2\text{Tr}(\text{Re}[(U_n^D)^\dagger U_n^O]), \qquad (24)$$

where equality is achieved with the minimum defect, as $U_n^O = U_n^D$. Because the left side of Eq. (24) is positive, the definition of fidelity is $F(U_n^D, U_n^O)$ in Eq. (2) in the main text.

### Photonic deep neural networks

To analyze the effect of defective unitaries on deep learning, we examine traditional supervised feed-forward neural networks with the error backpropagation method[34,35]. In the forward propagation, the

signal column vector of each layer $\Psi^p$ is updated with:

$$\Psi^{p+1} = W^p h_p(\Psi^p), \qquad (25)$$

where $h_p(\Psi^p)$ denotes the application of the activation function to each component of $\Psi^p$ through the computer-assisted simulation using electro-optic conversion[4]. We apply the tangent hyperbolic activation function[49] in hidden layers ($p = 2$ and $3$) and the linear activation function in the input ($p = 1$) and output ($p = 4$) layers.

The training and test datasets are obtained with the forward propagation of the neural network using the predefined weight matrices $W^p$. The elements of $W^p$ are obtained with the uniform random distribution $u[-1/M_p^2, 1/M_p^2]$. The datasets are then achieved with a set of the input vectors $\Psi^1$ obtained from $u[0,1]$ and its application to the predefined neural network. The training and test datasets consist of 4000 and 1000 pairs of input and output realizations, respectively.

The error backpropagation for network training is defined with the following equation[35]:

$$\Delta^p = h'(\Psi^p) \circ [(W^p)^T \Delta^{p+1}], \qquad (26)$$

where $h'(\Psi^p)$ denotes the derivative of the activation function and $\circ$ is the Hadamard product. We utilize the mean square error (MSE) as the loss function for updating the weight parameters with the output-layer error vector $\Delta$[4]. The weight matrices are updated with the mini-batch gradient descent (MGD) method[50] by dividing the training dataset into four mini-batches. The MGD leads to the following updating rules:

$$W^p(\tau_B + 1, \tau_E) = W^p(\tau_B, \tau_E) - \eta \langle (h'(\Psi^p) \circ [(W^p)^T \Delta^{p+1}])[h(\Psi^p)]^T \rangle, \qquad (27)$$

$$W^p(0, \tau_E + 1) = W^p(4, \tau_E), \qquad (28)$$

where $W^p(\tau_B, \tau_E)$ is the weight matrix between the $p$th and $(p+1)$th layers at the $(\tau_E)$th epoch with applying $\tau_B$ mini-batches, $\eta = 2 \times 10^6$ is the learning rate and $\langle \cdots \rangle$ is the average of the loss-function gradient for each mini-batch of the training dataset. After conducting Eq. (27) for a mini-batch, we apply Eq. (25) and recalculate the MSE loss function to employ the next mini-batch or epoch to Eq. (27). Starting from the initially random $W^p$ with $u[-1/M_p^2, 1/M_p^2]$, we train the neural network for 300 epochs. At each epoch, the loss function of the MSE for complex numbers is estimated with the test dataset to obtain the learning curves in Fig. 5b. The regression estimator of the R-squared is calculated with the test dataset after 300 epochs of training to obtain the accuracy curves in Fig. 5c.

## Data availability
Data used in the current study are available from the corresponding authors upon request and can also be obtained by running the shared codes at https://doi.org/10.24435/materialscloud:gj-y4 in the Materials Cloud Archive[51].

## Code availability
Codes used in this work are available at https://doi.org/10.24435/materialscloud:gj-y4 in the Materials Cloud Archive[51]. Supplementary Code 1 for Figs. 2–5 are available with this paper.

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

## Acknowledgements
We acknowledge financial support from the National Research Foundation of Korea (NRF) through the Basic Research Laboratory (No. 2021R1A4A3032027), Young Researcher Program (No. 2021R1C1C1005031) and Global Frontier Program (No. 2014M3A6B3063708), all funded by the Korean government. We also acknowledge an administrative support from SOFT foundry institute.

## Author contributions
S.Y. and N.P. conceived the project idea, discussed the results, and wrote the manuscript.

## Competing interests
The authors have no conflicts of interest to declare.
