## [Peer Review File · Nature Communications]

Heavy tails and pruning in programmable photonic circuits for universal unitariesREVIEWER COMMENTS

Reviewer #1 (Remarks to the Author):

The authors present a theoretical modelling and numerical testing of a stochastic large-scale programmable photonic circuits for the development of high-fidelity universal unitaries. To do so, they demonstrate that some elements such as MZI in photonic circuit are more important than phase shifters.

The paper is clearly written. Though, the implementation of pruning in an experimental system seems challenging and difficult to assess the practical future if this model.

Minor changes are requested:

1. The title is very general and seems to announce a review article and might be changed to be more specific and aligned with the targeted results.
2. The conclusion of abstract mentioned a universal architecture and this statement seems exaggerated. Indeed, their model applies specific cases of integrated circuits composed of phase shifter and MZI.
3. From which point do they define the tails? Does it significantly change the outcome if you modify the starting point of the tail distribution?
4. 'The observed heavy-tailed distribution signifies that some building blocks are more critical than others' This observation is not so clear to the reader, can they be more accurate or explain in more details based on the graphs they are showing?
5. How do they envision the realization of such an integrated photonic chip and his measurements? They should discuss other type of losses in such systems

Reviewer #2 (Remarks to the Author):

In this paper, the authors discovered a nontrivial stochastic property of large-scale programmable photonic circuits, i.e., heavy-tailed distributions of rotation operators. Programmable photonic circuits are promising candidates for universal unitary which is crucial for implementing quantum computations and deep learning accelerations. With increasing of circuit scales, the effects of noises become more significant and could degrade the performance of the system. Here, the authors demonstrated that rotation operators possess heave-tailed distributions, using a novel connection between large-scale photonic circuits and complex network systems. In particular, the authors revealed the presence of hub phase shifters, which allows for the application of network pruning to the design of photonic hardware that significantly suppresses the noises. Specifically, the authors developed a universal architecture for pruning random unitary matrices to achieve high fidelity and energy efficiency. Their results and discoveries have important implications for achieving high fidelity in large-scale quantum computing and photonic deep learning accelerators.

This paper is very well-written, with a very clear and concise background introduction of role of hardware development for high-dimensional unitary operators in implementing quantum computations and deep learning accelerations and the state-of-the-art research in programmable photonic circuits. It is of great topical interest and technically sound. Therefore, I would love to recommend its publication in Nat. Comm. Below are a couple of suggestions for optional minor revisions:

One of the key insights the authors obtained is that "some subsystems are more important than others, even in the highly symmetric architecture of large-scale programmable photonic circuits". It would be helpful if the authors could briefly elaborate this, e.g., what features are these more important sub-system possess (maybe in terms of network-based metrics)?

In Fig.2 Panel e, f, g, it would be helpful to directly put legends explaining the symbols in the plots. Also, the data plots also look a little fuzzy probably due to the loss of resolution when converting to PDF.

Reviewer #3 (Remarks to the Author):

The authors show that the pruning can maintain the fidelity of programmable photonic circuits to a certain degree. By reducing the number of phase modulators required, it is expected that the loss and noise can be improved.

Pruning has been a highly successful technique in digital neural networks. The authors took the idea to photonic circuits. The analysis is interesting and the results are well supported. However, the impact of pruning might be far less significant in photonic circuits. Neural networks use statistical learning, and it has built-in tolerance over fidelity of the output. For example, the classification performance is barely impacted by the precision of numerical output values. However, photonic circuits are often used as a coherent element. Small differences in output phase and amplitude caused by pruning could have a significant impact on the final output. If it is used as an incoherent element that will be followed by electric-optical conversion, maybe it is less of an issue. Maybe the authors can further elaborate its potential impact. In particular, evaluating fidelity as defined by the authors is not as compelling as demonstrating the performance degradation when pruned circuits are used for photonic neural computing.

The paper is interesting. But at its current form, it is hard to evaluate the impact. The fidelity as defined in the paper may not suffer from pruning. But that does not mean applications built on top of such photonic circuits will not suffer. Since small phase/amplitude differences can lead to huge differences as a result of coherent interference for downstream devices.

Reply to Reviewer 1's report

The authors present a theoretical modelling and numerical testing of a stochastic large-scale programmable photonic circuits for the development of high-fidelity universal unitaries. **To do so, they demonstrate that some elements such as MZI in photonic circuit are more important than phase shifters.**

The paper is clearly written. **Though, the implementation of pruning in an experimental system seems challenging and difficult to assess the practical future if this model.**

We sincerely appreciate the reviewer's careful and constructive comments on our manuscript. Following the reviewer's suggestions, we thoroughly revised the manuscript, and the quality of the manuscript was greatly improved, especially regarding practical realizations.

For the practical issue, we cordially emphasize that our manuscript focuses on removing a part of elements (here, phase shifters) that are relatively less important in the original design, including the works already demonstrated experimentally. For example, when we consider one of the state-of-the-art realizations of experimentally demonstrated programmable photonic circuits [*IEEE hot chips 32 symposium (HCS)* 1-26 (2020); *Light Sci. Appl.* **11**, 30 (2022)] which allows for $n = 64$ matrix multiplications, the system has required 2016 ($= n(n - 1)/2$) unit cells composed of 4032 MZIs and phase shifters. **The main claim of our work is to remove relatively less important phase shifters in the experimental platform because phase shifters have been one of the critical noise sources in programmable photonic circuits, as demonstrated in the reply to Point 5.**

In detail, if we assume about 5K temperature change of programmable circuits using silicon photonics, which may originate from the thermal crosstalk in the modulation in phase shifters, **the analysis in Fig. 4d of our manuscript shows that it is more advantageous to remove 50% (or 1008) θ -phase shifters in realizing U(64) circuits.** Figure 4d also demonstrates that this ratio of pruning becomes higher for larger n , which is strongly required for quantum computation and deep learning. We included the detailed discussion and revision in the reply to Point 5 in this reply letter.

Minor changes are requested:

1. The title is very general and seems to announce a review article and might be changed to be more specific and aligned with the targeted results.

Thank you very much for this valuable comment. We acknowledge that our previous title was overly general and did not accurately convey the specific focus of our work: analysing statistical properties and developing pruning techniques for programmable photonic circuits **aimed at achieving a universal form of unitary operators. We changed the title within the length limit of Nature Communications:**

(Title) Heavy tails and pruning in programmable photonic circuits for universal unitaries

2. The conclusion of abstract mentioned a universal architecture and this statement seems exaggerated. Indeed, their model applies specific cases of integrated circuits composed of phase shifter and MZI.

While our analysis presented in Figs. 2-4 can be applied to other designs (e.g., the Reck design) or to other optical elements (e.g., the unitary photonic circuit composed of the other forms of beam splitters and phase shifters), **we recognized the need to convey our results more accurately because these alternative designs may have different forms of universal architecture and some other forms of $U(n)$ decomposition methods may yield different statistical properties.** We therefore revised the sentence at the conclusion of abstract as follows:

(Abstract) For the Clements design of programmable photonic circuits, we extract a universal architecture for pruning random unitary matrices ...

3. From which point do they define the tails? Does it significantly change the outcome if you modify the starting point of the tail distribution?

Thank you for this excellent comment. **We defined the tail of each distribution according to the standard approach in statistics: the use of the maximum likelihood estimators (MLEs) and Kolmogorov–Smirnov (KS) test [SIAM Rev. 51, 661-703 (2009)], as described in the Methods section.** For all the statistical models, the maximization of the log-likelihood L with $\nabla L = 0$ determines the optimized parameters of an assumed distribution model for a given dataset. In the models having the explicit tail—the power-law, power-law with an exponential cutoff, and exponential distribution models—the cumulative distribution functions (CDF) of the models of each θ_{\min} are compared with the CDF of the dataset through the distance D in Eq. (22), which is called the KS test. This approach is the standard procedure for analysing heavy-tailed distributions.

The modification of the starting point of the tail will change the results, as the reviewer stated. **However, this modification should degrade the fitting quality of the model to a given dataset. In this context, our approach in deciding the starting point of the tail corresponds to the tail based on the optimized model for a given dataset.** We included further discussion for this point and more detailed explanations for the procedure in the revised manuscript.

(Lines 155-156) This standard procedure determines the range and shape of the tail of each model for the optimized fitting of a given dataset.

(Lines 402-404) ... we employ an analytical or numerical calculation of the maximum likelihood estimators (MLEs)²⁵ from the PDF $p(\vartheta_m; \{q_s\})$, which defines the probability of finding ϑ_m with the model having the parameters $\{q_s\}$.

(Lines 407-408) The model with the higher likelihood then provides the better fit to the dataset $\{\vartheta_m\}$ ¹⁹.

(Lines 476-477) This optimum model has the tail for the best fitting of a given dataset within the definition of each distribution.

4. ‘The observed heavy-tailed distribution signifies that some building blocks are more critical than others’ This observation is not so clear to the reader, **can they be more accurate or explain in more details based on the graphs they are showing?**

We are sorry for the confusion, and thank you for the nice suggestion. We revised the sentence with the description in Fig. 1e for more clarity.

(Lines 190-192) The observed heavy-tailed distribution of ϑ rotation operators signifies that some building blocks T_m^j with ϑ in the ‘Tail’ part in Fig. 1e are more critical than other building blocks (‘Body’ part in Fig. 1e).

5. How do they envision the realization of such an integrated photonic chip and his measurements?
They should discuss other type of losses in such systems

Thank you very much for this comment. We acknowledge that more extended discussion is necessary to clarify the impact of our work in relation to real implementation.

In the revised manuscript, **we clarified that the state-of-the-art realization of programmable photonic circuits using MZIs and phase shifters enables $n = 64$ operations. It is very supportive because this number fits well with the regime of excellent fitting with the power-law model ($n \geq 80$). Furthermore, as shown in Fig. 4d, the effect of pruning is also evident even in smaller numbers of $n = 16$ and 32 depending on the noise level.**

(Lines 276-281) Furthermore, let us consider one of the state-of-the-art realizations of experimentally demonstrated programmable photonic circuits, which allows for $n = 64$ matrix multiplications^{30,31}. The realization then requires $2016 (= n(n-1)/2)$ unit cells composed of 4032 MZIs and 2016 ϑ - and φ -phase shifters for unitary operations. When we consider thermal noise from about 5K temperature change, Fig. 4d shows that it is more advantageous to remove 50% (or 1008) ϑ -phase shifters in realizing U(64) circuits.

(Lines 352-354) It is worth mentioning that the state-of-the-art realization of programmable photonic circuits using MZIs and phase shifters allows the $n = 64$ matrix multiplication^{30,31}, which is close to the heavy-tailed regime.

We also discussed more details about the levels of noise in phase shifters, which are assumed in our work. We consider an example of thermo-optical phase shifters widely used in programmable photonic circuits [27]. This revision proves that the pruning method removing the source of noises enables superior performance of programmable photonic circuits.

(Lines 207-212) For example, consider a typical thermo-optic phase shifter with a device length of $100 \mu\text{m}$ ²⁷, which operates at the telecom wavelength of 1550 nm and is based on silicon photonics technology. The amount of thermal noise present in the phase evolution is determined by the thermo-optic coefficient of silicon²⁸ $dn/dT = 1.8 \times 10^{-4} \text{ K}^{-1}$, which can approach 0.02π per kelvin. This noise may be further exacerbated in larger-scale devices due to increasing thermal crosstalk.

(Figure 3 Caption) Two pairs of groups with noisy bodies and noisy tails are shown for $\delta_0 = 0.04\pi$ and 0.08π , which correspond to about $2\text{K} \sim 4\text{K}$ temperature changes in silicon infrared thermo-optical phase shifters²⁷.

(Figure 4 Caption) Two pairs of groups with noisy bodies and noisy tails are shown for $\delta_0 = 0.10\pi$ and 0.20π , which correspond to about $5\text{K} \sim 10\text{K}$ temperature changes in silicon infrared thermo-optical phase shifters²⁷.

Following the reviewer’s suggestion, we also provided the extended discussion on practical realizations in the revised manuscript, focusing on other types of loss.

(Lines 361-370) Although we studied the performance of pruning in photonic deep neural networks, various issues still remain for practical realizations. First, although we focused on unwanted phase shifts that may originate from thermal crosstalk, optical absorption from material loss or imperfect device fabrication may be

worse in large-scale photonic neural networks, especially when using gain or loss media⁴³. Second, the effect of other forms of nonlinear activation functions⁴⁴ and network architectures⁴⁵ on pruning performance should be studied. Finally, the statistical properties of weight matrices depend on problems, model architectures, learning methods, and hyperparameters^{46,47}. The resulting unitary matrices can have some biased distributions distinct from random Haar matrices, which will impose the problem- or model-specific properties in pruning performance.

Finally, in order to emphasize the usefulness of pruning in photonic neural networks, we included a new paragraph “Pruning in photonic deep neural networks” with Fig. 5. Learning curves (Fig. 5b) and accuracies (Fig. 5c) demonstrate that the pruning body allows for much more robust network learning than all the other cases of defective unitaries. This result verifies the usefulness of pruning in high-level photonic functionalities.

Fig. 5. Pruning in photonic deep neural networks for regression. **a**, The architecture of the deep neural network for analysing the effect of pruning. M_p ($p = 1, 2, 3, 4$) is the number of neurons in the p th layer, where $M_1 = M_4 = 16$ and $M_2 = M_3 = 32$. $\Psi^p = [a_1^p, a_2^p, \dots, a_{M_p}^p]^T$ and $\Delta^p = [\delta_1^p, \delta_2^p, \dots, \delta_{M_p}^p]^T$ are the signal and error column vectors at the p th layer, respectively, where a_k^p and δ_k^p are the signal and error at the k th neuron of each layer ($k = 1, 2, \dots, M_p$), respectively. W^p is the weight matrix between the p th and $(p+1)$ th layers. **b,c**, Learning curves (**b**) and R-squared accuracies (**c**) of the model, which are estimated with the test dataset. The colours of the lines denote the unitary photonic circuits in different groups: pruning body (red), pruning tail (blue), noisy body (orange), and noisy tail (green). In **b**, solid and dashed lines denote 5% ($2\sigma/n(n-1) = 0.05$) and 10% ($2\sigma/n(n-1) = 0.10$) defect ratios, respectively. In **c**, dashed lines and error bars represent the mean value and half of the standard deviation of the test dataset accuracies of 1,000 realizations. The disconnection of the blue dashed line in **c** denotes the failure of the training due to the divergence of network parameters. All the other parameters for training and calculating loss functions are shown in Methods.

Reply to Reviewer 2's report

In this paper, the authors discovered a nontrivial stochastic property of large-scale programmable photonic circuits, i.e., heavy-tailed distributions of rotation operators. Programmable photonic circuits are promising candidates for universal unitary which is crucial for implementing quantum computations and deep learning accelerations. With increasing of circuit scales, the effects of noises become more significant and could degrade the performance of the system. **Here, the authors demonstrated that rotation operators possess heave-tailed distributions, using a novel connection between large-scale photonic circuits and complex network systems.** In particular, the authors revealed the presence of hub phase shifters, which allows for the application of network pruning to the design of photonic hardware that significantly suppresses the noises. Specifically, the authors developed a universal architecture for pruning random unitary matrices to achieve high fidelity and energy efficiency. **Their results and discoveries have important implications for achieving high fidelity in large-scale quantum computing and photonic deep learning accelerators.**

This paper is very well-written, with a very clear and concise background introduction of role of hardware development for high-dimensional unitary operators in implementing quantum computations and deep learning accelerations and the state-of-the-art research in programmable photonic circuits. It is of great topical interest and technically sound. Therefore, I would love to recommend its publication in Nat. Comm. Below are a couple of suggestions for optional minor revisions:

We sincerely appreciate the reviewer's excellent summarization and positive comments on the novelty and impact of our work. **In this revision, we carefully revised the manuscript following the reviewer's suggestions, which helped improve the interdisciplinary understanding of our work and presentation of our manuscript.**

One of the key insights the authors obtained is that **"some subsystems are more important than others, even in the highly symmetric architecture of large-scale programmable photonic circuits"**. It would be helpful if the authors could briefly elaborate this, e.g., **what features are these more important sub-system possess (maybe in terms of network-based metrics)?**

Thank you very much for this comment. **We believe this is a very insightful comment pointing out the analogy of wave physics in programmable photonic circuits and network science regarding heavy-tailed distributions.**

When we consider the emergence of heavy-tailed distributions in network science, the phase shift θ that follows the power-law model may correspond to the number (or weight degree) of network links. In this case, the probability density function in Fig. 2e inset, which counts the number of SU(2) units at θ , implies that **the SU(2) unit will correspond to the network node.** From the above analogy, heavy-tailed distributions show that hub SU(2) unit elements (or important sub-systems) in Fig. 4a will be connected with a large number (or large weight degree) of links. **Yet, such an approach will require a rigorous definition of the link between SU(2) units, which should cover wave interferences over**

the entire programmable photonic circuits. This is a fascinating topic for the complete analogy between wave physics and network science in the platform of photonic circuits, similar to our recent work in defining link weight for scattering systems [Nat. Comput. Sci. 3, 128-138 (2023)]. However, at this stage, we believe that the topic is beyond the scope of this work. **In the context of the above idea, we included more details about the condition of hub elements, the analogy between network science and wave physics, and future research directions,** as follows:

(Lines 375-382) Regarding the emergence of heavy-tailed distributions in network science⁴⁸, the phase shift ϑ , or the rotation $R_z^m(-2\vartheta)$, may correspond to the number or weight degree of links that network nodes possess. In this analogy, each SU(2) unit corresponds to a network node, and different design methodologies of the U(n) decomposition¹¹⁻¹³ imply a set of distinct network architectures for the same signal behaviour U(n). To complete this analogy between wave physics and network science, we can envisage a network metric that quantifies the connectivity between SU(2) units, which should consider the interference effect, as demonstrated in the network modelling of wave scattering systems⁴⁸.

In Fig.2 Panel e, f, g, it would helpful to directly put legends explaining the symbols in the plots. Also, the data plots also look a little fuzzy probably due to the loss of resolution when converting to PDF.

Thank you very much for these careful comments. In the revised manuscript, we included the legends inside the plots of Fig. 2e,f,g. Also, we are sorry for the low resolution of the figures in the previous submission. We attached high-quality figures in the resubmission.

Reply to Reviewer 3's report

The authors show that the pruning can maintain the fidelity of programmable photonic circuits to certain degree. By reducing the number phase modulated required, it is expected the loss and noise can be improved.

Pruning has been a highly successful technique in digital neural network. The authors took the idea to photonic circuits. **The analysis is interesting and the results are well supported.**

Thank you very much for providing a thoughtful and insightful review of our work, especially regarding the impact of pruning for neural networks. Following the reviewer's advice, we included extended and detailed analysis to demonstrate the usefulness of pruning in photonic deep neural networks. We believe that the revision significantly improved the quality of my manuscript.

However, the impact of pruning might be far less significant in photonic circuits. Neural networks use statistic learning, and **it has built-in tolerance over fidelity of the output.** For example, the classification performance is barely impacted by the precision of numerical output value. **However, photonic circuits is often used as a coherent element. Small difference in output phase and amplitude caused by pruning could have significant impact on the final output.** If it is used as an incoherent element that will be followed by electric-optical conversion, maybe it is less an issue. Maybe the authors can further elaborate its potential impact. **In particular, evaluating fidelity as defined by authors is not as compelling as demonstrating the performance degradation when pruned circuits are used for photonic neural computing.**

We appreciate the excellent comment pointing out the necessary revision of our manuscript in terms of the impact of pruning for high-level functionalities, especially neural computing.

In the original manuscript, we demonstrated that the pruning method—**removing the operation of less important elements that may lead to noises**—provides superior fidelity to the programmable photonic circuit with noises. **In quantum applications using programmable photonic circuits, the fidelity itself is critical in achieving error-tolerant operations, as shown in applications in linear-optical quantum computing.** We included the related discussion in the revised manuscript.

(Lines 295-297) The importance of achieving high-fidelity photonic $U(n)$ operations demonstrated in Fig. 4 has been widely recognized in quantum computing, such as boson sampling¹ or quantum Fourier transform³² in linear-optical quantum computing (LOQC)³³.

However, as the reviewer commented precisely, it is not a simple issue in neural computing. **The fidelity of unitary operations is just a part of the entire functions of neural networks composed of possibly nonunitary weight matrices and nonlinear activation functions. Therefore, the analysis in Fig. 2-4 is insufficient to demonstrate the usefulness of pruning in neural network applications.** Further study needs to be carried out to confirm that the advantages obtained with pruning are still maintained even for high-level functionalities. We clarified this point in the revised manuscript.

(Lines 297-301) On the other hand, the validity of pruning in deep learning requires further clarification because

unitary operations are part of an entire neural network composed of weight matrices and activation functions^{4,34}. Therefore, we analyse how the pruning and adding noise in the unitary subparts of weight matrices impact the performance of photonic deep neural networks⁴.

To resolve this issue, we included a new section: **“Pruning in photonic deep neural networks” with the extended Methods section. In this new section, we explore the effect of fidelity in unitary operators on the learning and accuracy of photonic neural networks.** To focus on the impact of pruning, we investigated a traditional feedforward neural network and its realization with programmable photonic circuits [*Nat. Photon.* **11**, 441-446 (2017)]. **The idea is the comparison of learning performances when the unitary parts of weight matrices are pruned or have noises.** Notably, it is challenging to find universal results because many neural computing problems (e.g., MNIST, natural language processing, or image classification) handle a specific form of datasets and may possess some bias in network parameters [46,47]. **In this section, we set a regression problem defined with suppressed correlations: finding a specific form of a neural network (obtained from uniform random distributions) starting from another uniformly random network.** We included this discussion in the revised manuscript.

(Lines 302-309) To focus on the effect of pruning and adding noise, we consider a traditional feedforward neural network^{34,35} (Fig. 5a) with conventional training³⁶ and evaluation methods. The target problem is the regression of the relationship between the input and output datasets, which are connected through the neural network of Fig. 5a with a specific set of weight matrices $\{W^p | p = 1, 2, \text{ and } 3\}$. The goal of the network training starting from initially random weight matrices is the inference of the test output dataset from the test input, which corresponds to finding a specific form of a deep neural network from random deep neural networks (see Methods for details of the neural network model: architecture, datasets, training, hyperparameters, and loss function).

The critical result of this study is shown in Fig. 5b,c. **The learning curves of the regression in the assumed programmable photonic circuit (Fig. 5b) show that the pruning body (red solid and dashed lines) allows for much more robust network training than all the other cases of defective unitaries.** Noisy cases present relatively unstable learning, and the pruning tail results in the complete failure of learning. Such a tendency is also shown in the R-squared regression accuracy (R^2) after the training (Fig. 5c). Only the pruning body case provides an accuracy close to the ideal case without any defects (black solid line).

Fig. 5. Pruning in photonic deep neural networks for regression. **a**, The architecture of the deep neural network for analysing the effect of pruning. M_p ($p = 1, 2, 3, 4$) is the number of neurons in the p th layer, where $M_1 = M_4 = 16$ and $M_2 = M_3 = 32$. $\Psi^p = [a_1^p, a_2^p, \dots, a_{M_p}^p]^T$ and $\Delta^p = [\delta_1^p, \delta_2^p, \dots, \delta_{M_p}^p]^T$ are the signal and error column vectors at the p th layer, respectively, where a_k^p and δ_k^p are the signal and error at the k th neuron of each layer ($k = 1, 2, \dots, M_p$), respectively. W^p is the weight matrix between the p th and $(p+1)$ th layers. **b,c**, Learning curves (**b**) and R-squared accuracies (**c**) of the model, which are estimated with the test dataset. The colours of the lines denote the unitary photonic circuits in different groups: pruning body (red), pruning tail (blue), noisy body (orange), and noisy tail (green). In **b**, solid and dashed lines denote 5% ($2\sigma/n(n-1) = 0.05$) and 10% ($2\sigma/n(n-1) = 0.10$) defect ratios, respectively. In **c**, dashed lines and error bars represent the mean value and half of the standard deviation of the test dataset accuracies of 1,000 realizations. The disconnection of the blue dashed line in **c** denotes the failure of the training due to the divergence of network parameters. All the other parameters for training and calculating loss functions are shown in Methods.

We acknowledge that this result may not thoroughly analyse the usefulness of pruning in photonic neural networks. **For example, different types of loss functions (e.g., saturable functions accessible with optical nonlinearity), training methods (e.g., complex-valued neural network training), or network architecture (e.g., recurrent neural networks) may lead to the deviation from this result. However, we convince that the usefulness of pruning demonstrated in the most conventional form of artificial neural networks proves the impact of our work and shows the future research direction.** We included the related discussion in the revised manuscript.

(Lines 327-329) Although the case examined in Fig. 5b,c is a specific regression problem of a deep neural network model, the result shown in the same vein as Fig. 4 demonstrates the validity of the pruning method in photonic deep neural networks, serving as proof-of-concept at least.

(Lines 361-370) Although we studied the performance of pruning in photonic deep neural networks, various issues still remain for practical realizations. First, although we focused on unwanted phase shifts that may originate from thermal crosstalk, optical absorption from material loss or imperfect device fabrication may be worse in large-scale photonic neural networks, especially when using gain or loss media⁴³. Second, the effect of

other forms of nonlinear activation functions⁴⁴ and network architectures⁴⁵ on pruning performance should be studied. Finally, the statistical properties of weight matrices depend on problems, model architectures, learning methods, and hyperparameters^{46,47}. The resulting unitary matrices can have some biased distributions distinct from random Haar matrices, which will impose the problem- or model-specific properties in pruning performance.

The paper is interesting. But at its current form, it is hard to evaluate the impact. The fidelity as defined in the paper may not suffer from pruning. But that does not mean applications built on top of such photonic circuit will not suffer. **Since small phase/amplitude difference can lead to huge difference as a result of coherent interference for downstream devices.**

We appreciate the reviewer's insightful comment. We acknowledge that further studies will be essential to fully understand the impact of pruning in photonic neural computing. **However, our newly included section provides the analysis of phase incoherence accumulation from pruning because the studied work is based on the cascaded connections of 3 weight matrices with 6 unitary operations and the intermediate nonlinear activation functions.** In this environment, pruning body still persists its advantage with respect to other noisy or pruning tail cases. **To provide a more concrete evidence, we included further discussion on the plausibility of assumed noises with widely used thermo-optic phase shifters.**

(Lines 207-212) For example, consider a typical thermo-optic phase shifter with a device length of $100\ \mu\text{m}$ ²⁷, which operates at the telecom wavelength of 1550 nm and is based on silicon photonics technology. The amount of thermal noise present in the phase evolution is determined by the thermo-optic coefficient of silicon²⁸ $dn/dT = 1.8 \times 10^{-4}\ \text{K}^{-1}$, which can approach 0.02π per kelvin. This noise may be further exacerbated in larger-scale devices due to increasing thermal crosstalk.

(Figure 3 Caption) Two pairs of groups with noisy bodies and noisy tails are shown for $\delta_0 = 0.04\pi$ and 0.08π , which correspond to about $2\text{K} \sim 4\text{K}$ temperature changes in silicon infrared thermo-optical phase shifters²⁷.

(Figure 4 Caption) Two pairs of groups with noisy bodies and noisy tails are shown for $\delta_0 = 0.10\pi$ and 0.20π , which correspond to about $5\text{K} \sim 10\text{K}$ temperature changes in silicon infrared thermo-optical phase shifters²⁷.

By following the reviewer's comments, the quality of our manuscript is now greatly improved. Therefore, we cordially request the reconsideration of our revised manuscript.

[revised manuscript text omitted]
}\left(\text{Tr}\left[\left(U_n^D\right)^\dagger U_n^O\right]\right)}{n + \text{Tr}\left(\left(U_n^D\right)^\dagger U_n^D\right)}, \quad (2)$$

where U_n^O and U_n^D represent the original unitary matrix and its defective (pruned or noisy) one,
 respectively, and $\text{Tr}(A)$ is the trace of the square matrix A . Figure 3c shows the fidelities of each
 photonic circuit with the pruning or noise as a function of the ratio of defective elements: $2\sigma/n(n$
 $- 1)$ in the pruning groups. As expected, the fidelity is preserved much better when the body is
 pruned instead of the tail. More critical results are shown in comparison with the noisy circuits.
 When the noise amplitude increases, removing a specific ratio of the “body” phase shifters can be
 better for higher fidelity than the noisy ones, whether the noise is imposed on body or tail elements.
 Such a ratio, called the pruning threshold, increases with the noise level and scale of photonic
 circuits (Fig. 3d). This result states that there is a substantial restriction on the noise level in a
 large-scale programmable photonic circuit. If a phase shifter cannot meet this restriction, then it is
 better to remove the phase shifter to increase accuracy and decrease energy consumption for
 reconfigurability.

**Fig. 3. Pruning is often better than noise.** **a**, The concept of pruning in programmable photonic
 circuits. The phase shifter 2θ of the building block is replaced with an ordinary waveguide, which
 preserves the symmetry in the MZI arms. **b**, The noisy building block. The phase shifter 2θ is
 perturbed as $2(\theta + \delta)$. **c**, Comparison of the fidelities of the U_{128} photonic circuits in different
 groups: pruning body (red line), pruning tail (blue line), noisy body (orange line) and noisy tail
 (green error bars). The thicknesses of the coloured lines and the error bars present the range of the
 fidelities between their maxima and minima. The red arrows indicate the pruning thresholds for
 each case. Two pairs of groups with noisy bodies and noisy tails are shown for $\delta_0 = 0.04\pi$ and
 0.08π , which correspond to about $2\text{K} \sim 4\text{K}$ temperature changes in silicon infrared thermo-optical
 phase shifters²⁷. **d**, Pruning threshold as a function of the noise level δ_0 for different degrees of

unitary operators. In **c** and **d**, 100 random U_n realizations are analysed per value of n and defect
ratio.

**Universal architecture for pruning**

Although the result shown in Fig. 3 demonstrates hub functionality and the advantage of pruning
in realizing an individual unitary operator, it is insufficient to apply pruning to programmable
photonic circuits for universal unitary operators. This is because the sorted set Θ_n for pruning
varies with the form of a unitary operator. To apply the pruning method for universal unitaries
with reconfigurability, it is necessary to construct an adaptable architecture for the pruning process.

Because the position of each building block for nulling a specific off-diagonal element is
fixed in an n -degree photonic circuit, the averages of the phase rotations $\langle\theta_{m,l}\rangle$ and $\langle\varphi_{m,l}\rangle$ are
well-defined in hardware for random unitary operations that are uniformly sampled from $U(n)$.
Figures 4a and 4b describe the universal architectures defined by $\langle\theta_{m,l}\rangle$ and $\langle\varphi_{m,l}\rangle$, respectively,
for 100 U_n realizations of $n = 16$ and $n = 32$ (see Supplementary Note S6 for $n = 64$). As expected
from the distinct $SU(2)$ operations from θ and φ (Fig. 2b,c), we observe a spatially inhomogeneous
distribution of $\langle\theta_{m,l}\rangle$ in contrast to that of $\langle\varphi_{m,l}\rangle$. More specifically, the universal architectures
show significant θ -rotation contributions from the building blocks near the boundary of the
programmable photonic circuits. Such a consistent distribution allows for a universal sorted set
$\langle\Theta\rangle_n = \{\langle\theta_{m,l}\rangle_r | 1 \leq r \leq n(n-1)/2 \text{ for integer } r \text{ and } \langle\theta_{m,l}\rangle_p \leq \langle\theta_{m,l}\rangle_q \text{ for } p \leq q\}$ to develop a
pruning process applicable to any unitary operations.

From this guideline, we again employ pruning and add noise to the body and tail elements
to the set $\langle\Theta\rangle_n$. As shown in Figs 4c and 4d, the general tendencies in Figs
[revised manuscript text omitted]

 ≥ 0 , we obtain the relationship:

$$n + \text{Tr} \left((U_n^{\text{D}})^\dagger U_n^{\text{D}} \right) \geq 2 \text{Tr} \left(\text{Re} \left[(U_n^{\text{D}})^\dagger U_n^{\text{O}} \right] \right), \quad (24)$$

where equality is achieved with the minimum defect, as $U_n^{\text{O}} = U_n^{\text{D}}$. Because the left side of Eq.
 (24) is positive, the definition of fidelity is $F(U_n^{\text{D}}, U_n^{\text{O}})$ in Eq. (2) in the main text.

**Photonic deep neural networks.** To analyse the effect of defective unitaries on deep learning, we
 examine traditional supervised feed-forward neural networks with the error backpropagation
 method^{34,35}. In the forward propagation, the signal column vector of each layer Ψ^p is updated with:

$$\Psi^{p+1} = W^p h_p(\Psi^p), \quad (25)$$

where $h_p(\Psi^p)$ denotes the application of the activation function to each component of Ψ^p through
 the computer-assisted simulation using electro-optic conversion⁴. We apply the tangent hyperbolic
 activation function⁵⁰ in hidden layers ($p = 2$ and 3) and the linear activation function in the input
 ($p = 1$) and output ($p = 4$) layers.

The training and test datasets are obtained with the forward propagation of the neural network
 using the predefined weight matrices W^p . The elements of W^p are obtained with the uniform
 random distribution $u[-1/M_p^2, 1/M_p^2]$. The datasets are then achieved with a set of the input vectors
 Ψ^1 obtained from $u[0, 1]$ and its application to the predefined neural network. The training and test
 datasets consist of 4,000 and 1,000 pairs of input and output realizations, respectively.

The error backpropagation for network training is defined with the following equation³⁵:

$$\Delta^p = h'(\Psi^p) \circ \left[(W^p)^T \Delta^{p+1} \right], \quad (26)$$

where $h'(\Psi^p)$ denotes the derivative of the activation function. We utilize the mean square error
 (MSE) as the loss function for updating the weight parameters with the output-layer error vector
 Δ^4 . The weight matrices are updated with the mini-batch gradient descent (MGD) method⁵¹ by
 dividing the training dataset into 4 mini-batches. The MGD leads to the following updating rules:

$$W^p(\tau_B + 1, \tau_E) = W^p(\tau_B, \tau_E) - \eta \left\langle \left(h'(\Psi^p) \circ \left[(W^p)^T \Delta^{p+1} \right] \right) \left[h(\Psi^p) \right]^T \right\rangle, \quad (27)$$

$$W^p(0, \tau_E + 1) = W^p(4, \tau_E), \quad (28)$$

where $W^p(\tau_B, \tau_E)$ is the weight matrix between the p th and $(p+1)$ th layers at the (τ_E) th epoch with
 applying τ_B mini-batches, $\eta = 2 \times 10^6$ is the learning rate, \circ is the Hadamard product and $\langle \dots \rangle$ is the
 average of the loss-function gradient for each mini-batch of the training dataset. After conducting
 Eq. (27) for a mini-batch, we apply Eq. (25) and recalculate the MSE loss function to employ the
 next mini-batch or epoch to Eq. (27). Starting from the initially random W^p with $u[-1/M_p^2, 1/M_p^2]$,
 we train the neural network for 300 epochs. At each epoch, the loss function of the mean square
 error for complex numbers is estimated with the test dataset to obtain the learning curves in Fig.
 5b. The regression estimator of the R-squared is calculated with the test dataset after 300 epochs
 of training to obtain the accuracy curves in Fig. 5c.

**Data availability**

All data needed to evaluate the conclusions of this work are included in the paper and
 Supplementary Information. All raw data generated during the current study are available from the
 corresponding authors upon request or by running the shared codes at
 <https://doi.org/10.24435/materialscloud:gj-y4> in the Materials Cloud Archive⁵².

**Code availability**

Core codes developed in this work have been deposited at
 <https://doi.org/10.24435/materialscloud:gj-y4> in the Materials Cloud Archive⁵². The codes were

developed with MATLAB R2021a. Supplementary codes for Figs. 2–5 are available with this
paper.

**Acknowledgements**

We acknowledge financial support from the National Research Foundation of Korea (NRF)
through the Basic Research Laboratory (No. 2021R1A4A3032027), Young Researcher Program
(No. 2021R1C1C1005031) and Global Frontier Program (No. 2014M3A6B3063708), all funded
by the Korean government.

**Author contributions**

Both authors conceived the idea, discussed the results and contributed to the final manuscript.

**Competing interests**

The authors have no conflicts of interest to declare.

**Additional information**

**Correspondence and requests for materials** should be addressed to S.Y. or N.P.

**Reprints and permission information** is available at <http://www.nature.com/reprints>

**Figure Legends**

[revised manuscript text omitted]

**References**

- 1. Carolan, J., Harrold, C., Sparrow, C., Martín-López, E., Russell, N. J., Silverstone, J. W.,
Shadbolt, P. J., Matsuda, N., Oguma, M. & Itoh, M. Universal linear optics. *Science* **349**,
711-716 (2015).
- 2. Wang, J., Paesani, S., Ding, Y., Santagati, R., Skrzypczyk, P., Salavrakos, A., Tura, J.,
Augusiak, R., Mančinska, L. & Bacco, D. Multidimensional quantum entanglement with
large-scale integrated optics. *Science* **360**, 285-291 (2018).
- 3. Arrazola, J., Bergholm, V., Brádler, K., Bromley, T., Collins, M., Dhand, I., Fumagalli, A.,
Gerrits, T., Goussev, A. & Helt, L. Quantum circuits with many photons on a programmable
nanophotonic chip. *Nature* **591**, 54-60 (2021).
- 4. Shen, Y., Harris, N. C., Skirlo, S., Prabhu, M., Baehr-Jones, T., Hochberg, M., Sun, X.,
Zhao, S., Larochele, H. & Englund, D. Deep learning with coherent nanophotonic circuits.
*Nat. Photon.* **11**, 441-446 (2017).
- 5. Annoni, A., Guglielmi, E., Carminati, M., Ferrari, G., Sampietro, M., Miller, D. A., Melloni,
628 A. & Morichetti, F. Unscrambling light—automatically undoing strong mixing between
629 modes. *Light Sci. Appl.* **6**, e17110-e17110 (2017).
- 6. Klema, V. & Laub, A. The singular value decomposition: Its computation and some
applications. *IEEE Trans. Automat. Contr.* **25**, 164-176 (1980).

- 7. Wright, L. G., Onodera, T., Stein, M. M., Wang, T., Schachter, D. T., Hu, Z. & McMahon,
P. L. Deep physical neural networks trained with backpropagation. *Nature* **601**, 549-555
(2022).
- 8. Bogaerts, W., Pérez, D., Capmany, J., Miller, D. A., Poon, J., Englund, D., Morichetti, F.
& Melloni, A. Programmable photonic circuits. *Nature* **586**, 207-216 (2020).
- 9. Harris, N. C., Carolan, J., Bunandar, D., Prabhu, M., Hochberg, M., Baehr-Jones, T., Fanto,
638 M. L., Smith, A. M., Tison, C. C. & Alsing, P. M. Linear programmable nanophotonic
processors. *Optica* **5**, 1623-1631 (2018).
- 10. Deng, H. & Khajavikhan, M. Parity–time symmetric optical neural networks. *Optica* **8**,
1328-1333 (2021).
- 11. Reck, M., Zeilinger, A., Bernstein, H. J. & Bertani, P. Experimental realization of any
discrete unitary operator. *Phys. Rev. Lett.* **73**, 58 (1994).
- 12. Clements, W. R., Humphreys, P. C., Metcalf, B. J., Kolthammer, W. S. & Walmsley, I. A.
Optimal design for universal multiport interferometers. *Optica* **3**, 1460-1465 (2016).
- 13. Saygin, M. Y., Kondratyev, I., Dyakonov, I., Mironov, S., Straupe, S. & Kulik, S. Robust
architecture for programmable universal unitaries. *Phys. Rev. Lett.* **124**, 010501 (2020).
- 14. Bouland, A. & Aaronson, S. Generation of universal linear optics by any beam splitter.
*Phys. Rev. A* **89**, 062316 (2014).
- 15. Fldzhyan, S. A., Saygin, M. Y. & Kulik, S. P. Optimal design of error-tolerant
reprogrammable multiport interferometers. *Opt. Lett.* **45**, 2632-2635 (2020).
- 16. Chen, X., Stroobant, P., Pickavet, M. & Bogaerts, W. Graph representations for
programmable photonic circuits. *J. Lightwave Technol.* **38**, 4009-4018 (2020).
- 17. Barabási, A.-L. *Network science* (Cambridge university press, 2016).

- 18. Barabási, A.-L. & Bonabeau, E. Scale-Free Networks. *Sci. Am.* **288**, 60-69 (2003).
- 19. Clauset, A., Shalizi, C. R. & Newman, M. E. Power-law distributions in empirical data.
*SIAM Rev.* **51**, 661-703 (2009).
- 20. Broido, A. D. & Clauset, A. Scale-free networks are rare. *Nat. Commun.* **10**, 1017 (2019).
- 21. Voitalov, I., van der Hoorn, P., van der Hofstad, R. & Krioukov, D. Scale-free networks
well done. *Phys. Rev. Res.* **1**, 033034 (2019).
- 22. Hamerly, R., Bandyopadhyay, S. & Englund, D. Stability of self-configuring large
multiport interferometers. *Phys. Rev. Appl.* **18**, 024018 (2022).
- 23. Blalock, D., Gonzalez Ortiz, J. J., Frankle, J. & Gutttag, J. What is the state of neural
network pruning? *Proceedings of machine learning and systems* **2**, 129-146 (2020).
- 24. Haar, A. Der Massbegriff in der Theorie der kontinuierlichen Gruppen. *Ann. Math.*, 147-
169 (1933).
- 25. Hill, B. M. A simple general approach to inference about the tail of a distribution. *Ann.*
*Stat.*, 1163-1174 (1975).
- 26. Press, W. H., Teukolsky, S. A., Vetterling, W. T. & Flannery, B. P. *Numerical recipes 3rd*
*edition: The art of scientific computing* (Cambridge university press, 2007).
- 27. Zhang, H., Gu, M., Jiang, X., Thompson, J., Cai, H., Paesani, S., Santagati, R., Laing, A.,
Zhang, Y. & Yung, M. An optical neural chip for implementing complex-valued neural
network. *Nat. Commun.* **12**, 457 (2021).
- 28. Komma, J., Schwarz, C., Hofmann, G., Heinert, D. & Nawrodt, R. Thermo-optic
coefficient of silicon at 1550 nm and cryogenic temperatures. *Appl. Phys. Lett.* **101**, 041905
(2012).
- 29. Cabrera, R., Shir, O. M., Wu, R. & Rabitz, H. Fidelity between unitary operators and the

- generation of robust gates against off-resonance perturbations. *Jour. Phys. A* **44**, 095302
(2011).
- 30. Zhou, H., Dong, J., Cheng, J., Dong, W., Huang, C., Shen, Y., Zhang, Q., Gu, M., Qian, C.
& Chen, H. Photonic matrix multiplication lights up photonic accelerator and beyond. *Light*
*Sci. Appl.* **11**, 30 (2022).
- 31. Ramey, C. *Silicon photonics for artificial intelligence acceleration: HotChips 32*. In *2020*
*IEEE hot chips 32 symposium (HCS)* 1-26 (2020).
- 32. Bartlett, B. & Fan, S. Universal programmable photonic architecture for quantum
information processing. *Phys. Rev. A* **101**, 042319 (2020).
- 33. Kok, P., Munro, W. J., Nemoto, K., Ralph, T. C., Dowling, J. P. & Milburn, G. J. Linear
optical quantum computing with photonic qubits. *Rev. Mod. Phys.* **79**, 135 (2007).
- 34. LeCun, Y., Bengio, Y. & Hinton, G. Deep learning. *Nature* **521**, 436 (2015).
- 35. Bishop, C. M. & Nasrabadi, N. M. *Pattern recognition and machine learning* (Springer,
2006).
- 36. Hecht-Nielsen, R. in *Neural networks for perception* 65-93 (Elsevier, 1992).
- 37. Arute, F., Arya, K., Babbush, R., Bacon, D., Bardin, J. C., Barends, R., Biswas, R., Boixo,
S., Brandao, F. G. & Buell, D. A. Quantum supremacy using a programmable
superconducting processor. *Nature* **574**, 505-510 (2019).
- 38. Shastri, B. J., Tait, A. N., Ferreira de Lima, T., Pernice, W. H., Bhaskaran, H., Wright, C.
D. & Prucnal, P. R. Photonics for artificial intelligence and neuromorphic computing. *Nat.*
*Photon.* **15**, 102-114 (2021).
- 39. Zhang, H., Gu, M., Jiang, X., Thompson, J., Cai, H., Paesani, S., Santagati, R., Laing, A.,
Zhang, Y. & Yung, M. An optical neural chip for implementing complex-valued neural

- network. *Nat. Commun.* **12**, 1-11 (2021).
- 40. Preskill, J. Quantum computing in the NISQ era and beyond. *Quantum* **2**, 79 (2018).
- 41. Arrazola, J. M., Bergholm, V., Brádler, K., Bromley, T. R., Collins, M. J., Dhand, I.,
Fumagalli, A., Gerrits, T., Goussev, A. & Helt, L. G. Quantum circuits with many photons
on a programmable nanophotonic chip. *Nature* **591**, 54-60 (2021).
- 42. Steinbrecher, G. R., Olson, J. P., Englund, D. & Carolan, J. Quantum optical neural
networks. *Npj Quantum Inf.* **5**, 1-9 (2019).
- 43. Huang, C., Sorger, V. J., Miscuglio, M., Al-Qadasi, M., Mukherjee, A., Lampe, L., Nichols,
709 M., Tait, A. N., Ferreira de Lima, T. & Marquez, B. A. Prospects and applications of
710 photonic neural networks. *Advances in Physics: X* **7**, 1981155 (2022).
- 44. Miscuglio, M., Mehrabian, A., Hu, Z., Azzam, S. I., George, J., Kildishev, A. V., Pelton, M.
& Sorger, V. J. All-optical nonlinear activation function for photonic neural networks.
*Optical Materials Express* **8**, 3851-3863 (2018).
- 45. Qu, Y., Zhou, M., Khoram, E., Yu, N. & Yu, Z. Resonance for analog recurrent neural
network. *ACS Photon.* **9**, 1647-1654 (2022).
- 46. Martin, C. H. & Mahoney, M. W. *Heavy-tailed Universality predicts trends in test*
*accuracies for very large pre-trained deep neural networks*. In *Proceedings of the 2020*
*SIAM International Conference on Data Mining* 505-513 (2020).
- 47. Yu, S., Piao, X. & Park, N. Machine learning identifies scale-free properties in disordered
materials. *Nat. Commun.* **11**, 4842 (2020).
- 48. Yu, S. Evolving scattering networks for engineering disorder. *Nature Computational*
*Science* **3**, 128-138 (2023).
- 49. Yu, S., Qiu, C.-W., Chong, Y., Torquato, S. & Park, N. Engineered disorder in photonics.

- *Nat. Rev. Mater.* **6**, 226-243 (2021).
- 50. Sharma, S., Sharma, S. & Athaiya, A. Activation functions in neural networks. *Towards*
*Data Sci* **6**, 310-316 (2017).
- 51. Ruder, S. An overview of gradient descent optimization algorithms. *arXiv preprint*
*arXiv:1609.04747* (2016).
- 52. Yu, S. & Park, N. *Pruning photonic circuits for universal unitary operators* (Materials
Cloud Archive, 2023; <https://doi.org/10.24435/materialscloud:gj-y4>, 2023).

REVIEWERS' COMMENTS

Reviewer #1 (Remarks to the Author):

The authors addressed my comments in a satisfactory way.

Reviewer #2 (Remarks to the Author):

The authors have sufficiently addressed all of my comments and suggestions. The paper can be published in the current form.

Reviewer #3 (Remarks to the Author):

the paper has been improved by acknowledging the limitation. I can recommend for publication.

Reply to Reviewer 1's report

The authors addressed my comments in a satisfactory way.

We are deeply grateful for the reviewer's significant contributions during the peer review process and the recommendation for the revised manuscript.

Reply to Reviewer 2's report

The authors have sufficiently addressed all of my comments and suggestions. The paper can be published in the current form.

We sincerely appreciate the reviewer's significant efforts during the peer review process and recommendation for publication.

Reply to Reviewer 3's report

the paper has been improved by acknowledging the limitation. I can recommend for publication.

Thank you very much for your efforts during the peer review process. We sincerely appreciate the reviewer's recommendation for the publication of the revised manuscript.